# Loess deposits in the low latitudes of East Asia reveal the ~20-kyr precipitation cycle

Xusheng Li [1], Yuwen Zhou [1], Zhiyong Han [1] ✉, Xiaokang Yuan[1], Shuangwen Yi[1], Yuqiang Zeng[1], Lisha Qin[1], Ming Lu[1] & Huayu Lu [1]

The cycle of precipitation change is key to understanding the driving mechanism of the East Asian summer monsoon (EASM). However, the dominant cycles of EASM precipitation revealed by different proxy indicators are inconsistent, leading to the "Chinese 100 kyr problem". In this study, we examine a high-resolution, approximately 350,000-year record from a low-latitude loess profile in China. Our analyses show that variations in the ratio of dithionite−citrate−bicarbonate extractable iron to total iron are dominated by the ~20-kyr cycle, reflecting changes in precipitation. In contrast, magnetic susceptibility varies with the ~100-kyr cycle and may be mainly controlled by temperature-induced redox processes or precipitation-induced signal smoothing. Our results suggest that changes in the EASM, as indicated by precipitation in this region, are mainly forced by precession-dominated insolation variations, and that precipitation and temperature may have varied with different cycles over the past ~350,000 years.

As an important part of the global monsoon system, the Asian summer monsoon is divided into the Indian summer monsoon (ISM), the Northwest Pacific summer monsoon (WNPSM), and the East Asian summer monsoon (EASM)[1]. The formation of the modern Asian summer monsoon is linked to seasonal surface temperature gradients. As insolation increases in the Northern Hemisphere, the surface temperature gradient between the Asian continent and the Indo−Pacific Ocean increases, causing winds to blow towards the continent. Simultaneously, water vapor is transported to the continental interior, leading to the formation of Asian summer monsoon precipitation[2]. The Asian summer monsoon, which can generate abundant precipitation, plays a crucial role in maintaining the ecosystem of East Asia.

Studies of Chinese mid-latitude loess have produced many geological records of the EASM, of which those from the Chinese Loess Plateau (CLP) are the most classic[3–6]. Magnetic susceptibility (MS) in most cases reflects the dominant 100-kyr cycle of the EASM since the Middle Pleistocene[3,7–11]. Both δ[13]C of inorganic carbonate[12] and δ[13]C of land snail shells[13] exhibit the same cycle. Because the strong EASM recorded on the CLP occurred during the interglacial period, which was characterized by low global ice volume, the change in the EASM is thought to be driven by global ice volume variability[7,13]. Unlike the

proxies mentioned above, a spliced EASM record since 260 ka from the western CLP reveals a dominant 23-kyr cycle caused by Northern Hemisphere summer insolation forcing[14]; and a composite microcodium δ[18]O record since the last interglacial period from the central CLP shows precessional cycles[15].

Stalagmites in East Asia have also recorded the summer monsoon change since the Middle Pleistocene[16–19]. The stalagmite δ[18]O records exhibit remarkable spatial and temporal consistency at the orbital scale, and the oxygen isotope indicator in stalagmites is currently considered a proxy for summer monsoon intensity rather than precipitation[2,19]. In contrast to most loess records, the stalagmite δ[18]O records display a dominant ~20-kyr period for the summer monsoon, with almost no ~100-kyr periodic component. The stalagmite records suggest that a strong EASM corresponds to a period of high insolation in the Northern Hemisphere. Therefore, the EASM is thought to be driven by precessional processes[2,19].

The idea that the EASM is mainly driven by changes in insolation controlled by precession is supported by simulations[20,21]. Similar results have been obtained with different models, i.e., the southwesterlies and southeasterlies are strengthened and reach farther inland when insolation increases in the Northern Hemisphere, leading

[1]School of Geography and Ocean Science, Nanjing University, Nanjing 210023, China. ✉e-mail: zyhan@nju.edu.cn

to a northwards shift in the rainfall belt, an overall increase in continental precipitation[22] and negative $\delta^{18}O$ values for continental precipitation[23,24]. In this scenario, loess and stalagmite records should indicate the same precessional cycle; however, the dominant cycles of the EASM differ in these records, resulting in the so-called "Chinese 100-kyr problem"[2,19,25]. It has been suggested that the global monsoon should be reconciled with the existing framework of orbital theory[19].

It has been speculated that several factors may contribute to the occurrence of the "Chinese 100-kyr problem": (1) the unique geographical location of the CLP may play a role, as simulations have shown that the effect of precession on precipitation in this region is not significant[22–24]; (2) the MS does not include changes in deposition rates, and the 20-kyr cycle is nearly as important as the 100-kyr cycle in terms of the MS flux[26]; (3) loess MS is not a pure monsoon precipitation signal, but a measure of more processes than the intensity of the EASM alone[27]; (4) there may be some mechanisms that suppress the variability in the magnitude of stalagmite $\delta^{18}O$ at the glacial–interglacial scale[28] (for example, the moisture transport pathway effect may counteract the forcing of glacial boundary conditions[29], resulting in the lack of glacial–interglacial variability in the stalagmite $\delta^{18}O$ records); (5) post-depositional processes have smoothed the MS signal, increasing the influence of the ~100-kyr cycles and decreasing the influence of the ~20- and ~40-kyr cycles[30]; and (6) reduced precipitation during glacials on the CLP may have fallen below the threshold for magnetic enhancement, preventing MS from recording monsoon precipitation during glacials, although it remains recordable during interglacials[14].

To determine the cause of the "Chinese 100-kyr problem", it is useful to obtain geological records from locations distant from the CLP. Two records from the monsoon marginal zone show a wet–dry variation similar to that of the East Asian speleothem over the past ~300 kyr[14,31]. However, records from the monsoon core zone are still lacking. Therefore, the loess in the lower reaches of the Yangtze River was chosen as the subject of this study (Fig. 1). The loess distributed in this region, known as the Xiashu loess, began to accumulate ~0.9 Myr ago[32]. The loess in Madang, Pengze County, Jiangxi Province, has special characteristics: Madang is located at a low latitude (Fig. 1 and Supplementary Fig. 1), corresponding to the monsoon core zone, while the CLP is located at mid-latitudes, roughly corresponding to the monsoon marginal zone.

## Results

### Madang profile and loess stratigraphy

The Madang profile (29°59′29″N, 116°39′44″E), with a thickness of 15.9 m, is roughly divided into seven loess–paleosol stratigraphic units, which roughly correspond to the stratigraphic units L1, S1, L2, S2, L3, S3 and L4 of the CLP (Supplementary Fig. 3). A total of 291 samples were collected and analyzed for grain size, MS, frequency-dependent magnetic susceptibility (FDS), dithionite–citrate–bicarbonate (DCB) extractable iron (FeD), total iron (FeT), hematite content and redness. Additionally, six optically stimulated luminescence (OSL) samples were taken from units L1 and S1. The grain size analysis showed that the samples have typical frequency distribution curves for eolian sediments and do not vary much in the profile (Supplementary Fig. 4). The MS and FDS variations show a strong positive correlation (Supplementary Figs. 5 and 6), and their curves are similar to those of the CLP loess. The obtained OSL ages (Supplementary Table 4) are consistent with the stratigraphic sequence, ranging from 30.1 ± 2.2 ka (at 40 cm depth) to

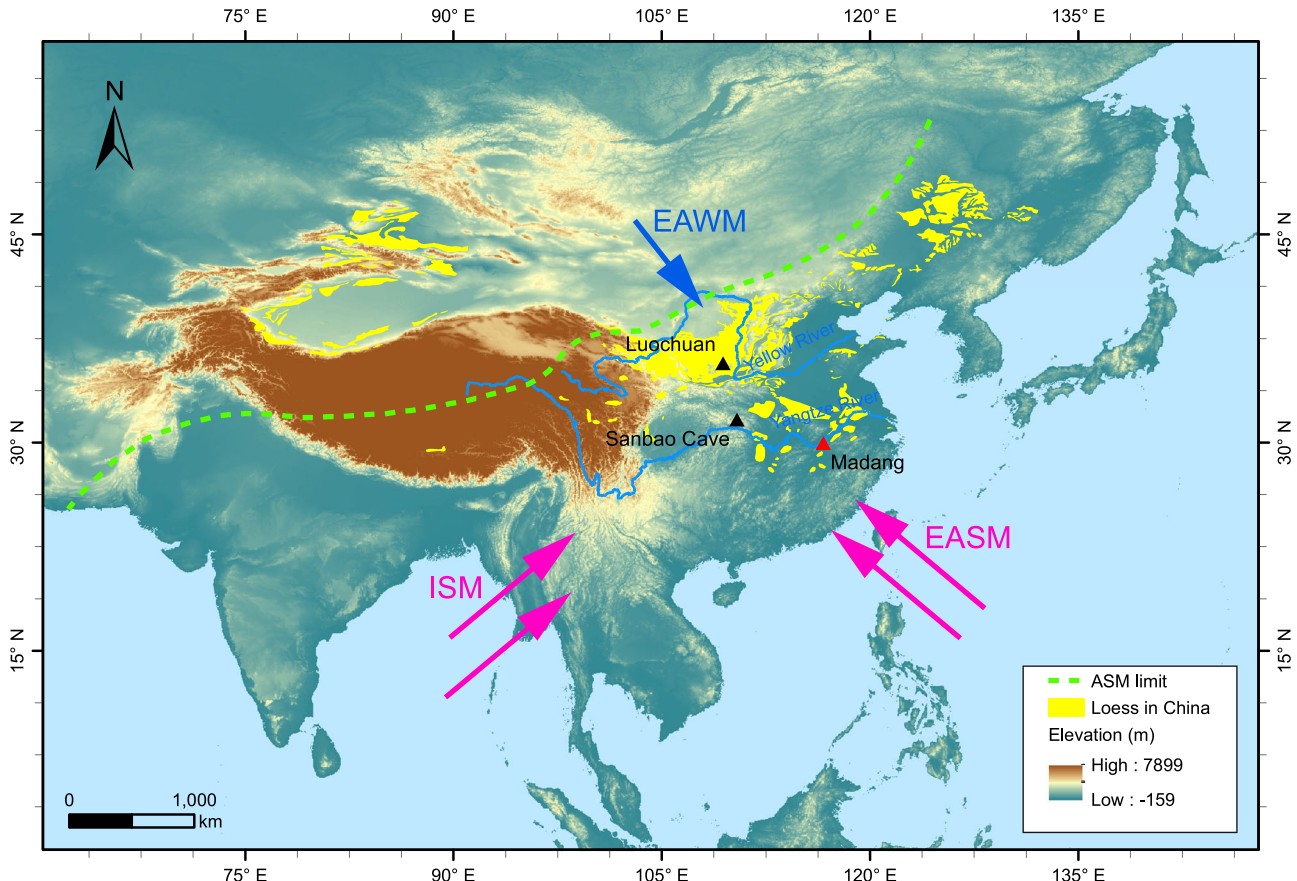

**Fig. 1 | Map showing the East Asian summer monsoon (EASM), the East Asian winter monsoon (EAWM), and the Indian summer monsoon (ISM).** The dashed green lines represent the modern Asian summer monsoon (ASM) limit[67]. The triangles represent the locations of the study (red) and reference (black) sites. A digital elevation model (SRTM V4)[68] used in the figure is available from the CGIAR-CSI SRTM 90 m Database (http://srtm.csi.cgiar.org).

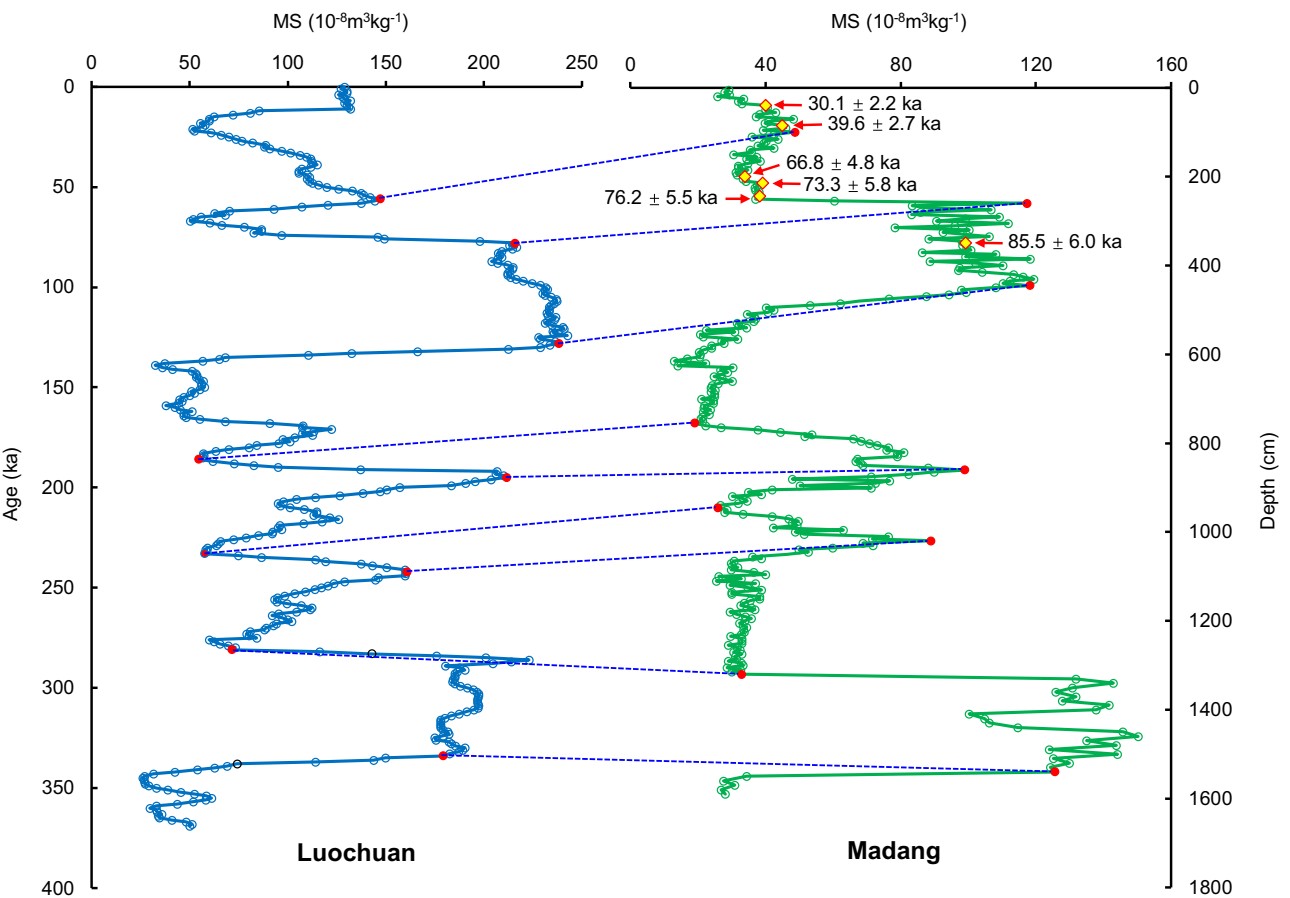

**Fig. 2 | Comparison of magnetic susceptibility curves between the Madang and Luochuan profiles**[8]. The tie points (red dots) are linked by dashed blue lines. The yellow diamonds represent the depths of the optically stimulated luminescence (OSL) samples. The OSL ages for samples at each depth are also shown.

$85.5 \pm 6.0$ ka (at 350 cm depth). The time series of MS defined by OSL ages can be well correlated with that of the Luochuan profile, a representative loess–paleosol sequence from the central CLP. In particular, the MS variation curves of the Madang and Luochuan profiles show striking similarities (Fig. 2). Combined with the OSL dating results, the Madang MS curve can be directly compared with that of Luochuan, with a corresponding time scale established based on the orbital tuning method[8]. Based on the nine depth–age tie points (Fig. 2 and Supplementary Table 5), the time scale of the Madang profile can be established by linear interpolation and extrapolation. This time scale shows that the bottom of the profile is 346.0 ka, the top is 42.9 ka (Fig. 2 and Supplementary Table 6), and the sampling resolution of the profile is approximately 1 kyr. Using a computational program for unevenly spaced time series[33], we estimated the cross-correlation coefficient ($r_{xy}$) between the Madang and Luochuan MS time series to be 0.88 (Supplementary Fig. 7), indicating a strong positive correlation.

### Variation cycles of MS and FeD/FeT

Wavelet analysis and power spectrum analysis show that there is a dominant 101 kyr cycle in the MS curve of the Madang profile (95% robust confidence intervals) (Fig. 3a, b). The ~100-kyr cycle is considered to be an eccentricity-dominated cycle[34], which also appears in the integrated MS curve of the CLP (0–350 ka)[35]. In contrast, there is a distinct 18 kyr cycle in the FeD/FeT curve (95% robust confidence intervals) (Fig. 3c, d). The 18 kyr cycle closely resembles the ~20-kyr precession cycle, suggesting that the FeD/FeT curve is generally dominated by a precession cycle. It is evident that the dominant cycle of the MS curve is significantly different from that of the FeD/FeT curve in the Madang profile.

## Discussion

Unlike the situation in the Madang profile, the dominant cycles of the MS and FeD/FeT curves for the CLP loess are essentially the same. The FeD/FeT changes since the middle Pleistocene show a clear 100-kyr dominant cycle and FeD/FeT is considered a good indicator of the summer monsoon[36,37]. The MS curves are well correlated with the deep-sea $\delta^{18}O$ curves, which exhibit a primary 100-kyr cycle since the middle Pleistocene, and are generally considered to represent changes in the EASM driven by the global ice volume[4,38–40].

The ratio of FeD/FeT, a widely used proxy indicator to evaluate the intensity of soil chemical weathering, is considered a measure of the quantity of iron liberated from iron-bearing silicate minerals relative to the total available iron[36,41]. The effects of precipitation and temperature on chemical weathering rates must be considered simultaneously[42]. A study of modern soils with different parent materials along a north–south climatic gradient in eastern China indicated positive relationships between FeD and mean annual precipitation (MAP) and between FeD and mean annual temperature (MAT). However, an obvious increase in FeD is only observed when the MAP exceeds ~800 mm (correspondingly, the MAT is higher than ~15 °C) (Supplementary Fig. 8)[43]. This may explain why changes in FeD/FeT occurred on the CLP without a noticeable precession cycle. The modern MAP and MAT on the CLP are relatively low, with only 592 mm and 9.6 °C, respectively, observed at Luochuan. Under high rainfall regimes, such as in southern China, the enrichment of FeD controlled by chemical weathering is favored by an increase in rainfall, as long as it does not exceed the specified rainfall threshold[44]. A study conducted in South China revealed a strong positive correlation between FeD/FeT and MAP when MAP ranged from 900 mm to 1720 mm[44] ($r = 0.83$;

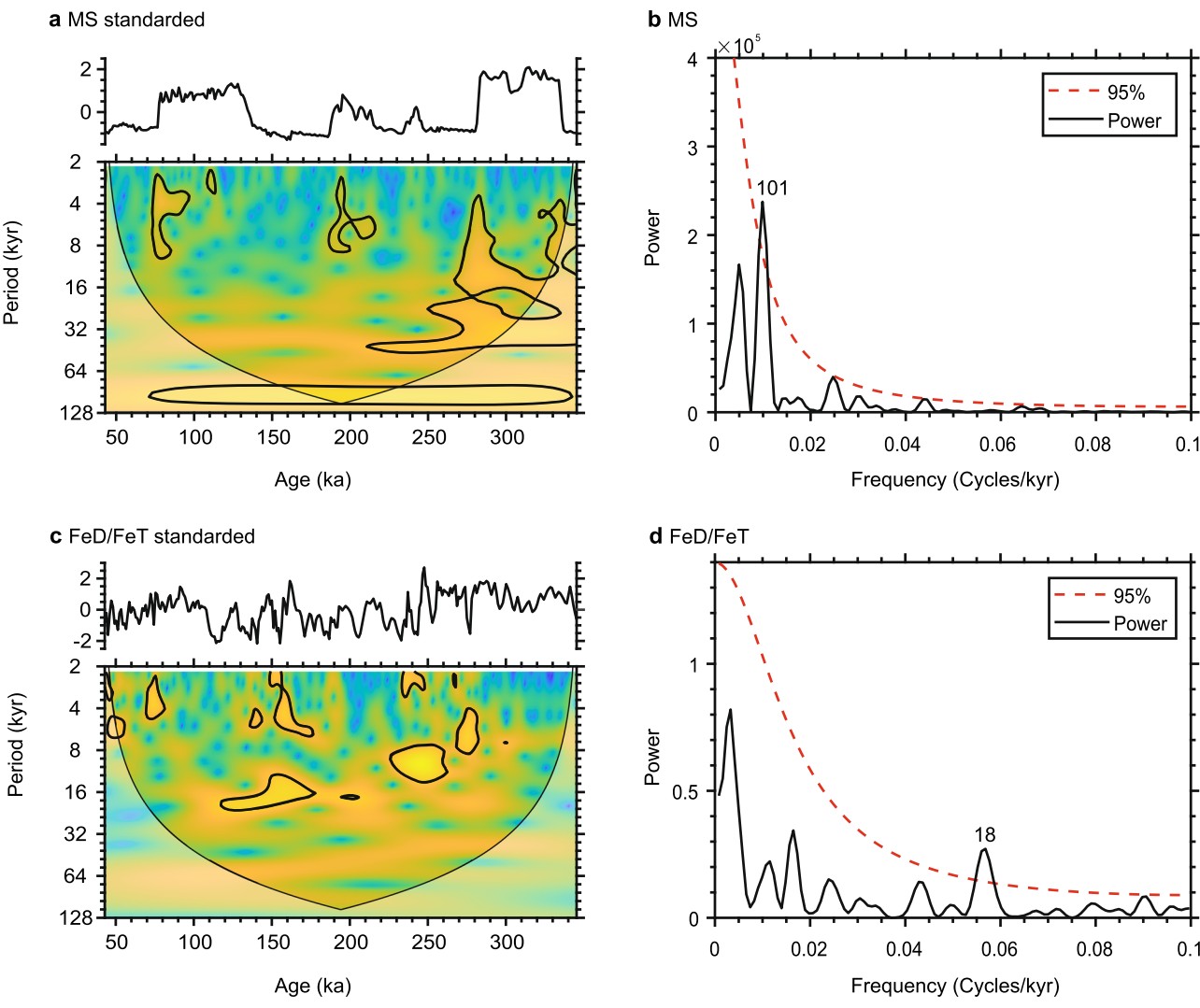

**Fig. 3 | Wavelet and spectral analysis results for the low-latitude loess records. a** Wavelet analysis of magnetic susceptibility. **b** Spectral analysis of magnetic susceptibility. **c** Wavelet analysis of FeD/FeT. **d** Spectral analysis of FeD/FeT.

Supplementary Fig. 9). However, no correlation was found between FeD/FeT and MAT in the study ($r = -0.08$; Supplementary Fig. 10). It is evident that FeD/FeT is a precipitation dominated proxy indicator in low latitude areas.

The increased MS (or FDS) of loess on the CLP has been shown to be related to pedogenesis[45]. The main climatic factors influencing pedogenesis are temperature and moisture. Studies in the CLP and beyond have shown that topsoil MS is statistically correlated with both temperature and precipitation[46,47]. Because temperature and precipitation gradients are spatially similar in the EASM region, it is difficult to separate the primary controlling factors based on topsoil studies. Although Luochuan and Madang are far apart (900 km) and are located in the monsoon marginal zone and monsoon core zone, respectively, the MS curves of these two sites are remarkably similar (Fig. 2). This suggests that MS reflects a climatic factor that varies consistently over a large region, or that these sites have undergone similar post-depositional processes.

Could both MS and FeD/FeT serve as proxies for precipitation in Madang? The existing evidence is insufficient to provide a definitive answer, but two hypotheses can be proposed: (1) Both MS and FeD/FeT serve as proxies for precipitation; (2) MS and FeD/FeT have different climatic indications.

The first hypothesis involves post-depositional processes. Previous study[30] suggested that leaching-related smoothing can diminish the strength of precessional cycles. Pedogenesis during interglacial periods can penetrate to greater depths, increasing the MS in underlying materials and attenuating the precessional signal in CLP loess deposits[30]. The MS and FeD/FeT records in the CLP show similar cycles, suggesting that smoothing has a similar influence on both proxies. In contrast, the MS and FeD/FeT records in Madang show different cycles, which would require precisely controlled differential smoothing to preserve the 20-kyr cycle of FeD/FeT but erase the 20-kyr cycle of MS. One possible interpretation is that the climatic conditions at the Madang site favor the production of FeD compared to the CLP[48]. Consequently, the FeD content at the Madang site significantly exceeds the concentration of ultrafine ferrimagnetic grains, which determines the degree of magnetic enhancement. As a result, the FeD/FeT proxy at the Madang site may be less influenced by leaching-induced signal smoothing, unlike the CLP site where DCB-soluble iron oxides may be roughly equivalent to ultrafine ferrimagnetic minerals.

The second hypothesis involves the influence of temperature. Precipitation may not be a controlling factor for MS enhancement at wet low latitudes. For example, the MS in Madang is generally lower than that in Luochuan. Excessive precipitation may suppress the enhancement of MS, as the MS of modern soils in southern China decreases with increasing precipitation[46]. A study has revealed an anomalous decrease in MS in the Xiashu loess, attributed to a strong reducing action caused by excessive soil moisture[49]. Soil moisture

decreases with increasing evapotranspiration, which mainly increases with temperature. In this sense, MS enhancement at the Madang site may be more sensitive to temperature, since changes in temperature lead to changes in redox status. The Madang MS curve compares well with the temperature proxy curve for an Antarctic ice core[50], revealing a clear positive correlation ($r_{xy} = 0.61$) (Fig. 4a). High MS values correspond to interglacial periods, while low MS values correspond to glacial periods (Fig. 4b). The cross-correlation coefficient between the MS of Madang and the benthic $\delta^{18}O$ records of the LR04 stack[51] is −0.70, showing a clear negative correlation. These correlations suggest that MS in Madang may primarily reflect variations in temperature.

The MS enhancement of loess at low latitudes is also related to local landforms. Unlike the MS curves of the Madang profile, those of other loess profiles in the lower reaches of the Yangtze River show inconsistencies when compared with those of Luochuan[32]. The lack of excessive soil moisture in the Madang profile may be due to the fast-draining and well-ventilated local topography (Supplementary Fig. 1). The Madang profile is situated in the piedmont zone along the Yangtze River, with a ground elevation exceeding 30 m. Simultaneously, the Yangtze River valley in Pengze County, where Madang is located, follows a northeast-southwest direction, in line with the prevailing local wind direction (Supplementary Fig. 1). The Madang profile benefits from relatively high wind speeds and its elevated position, contributing to excellent ventilation and drainage conditions.

Using an orbital-tuned age model is a widely adopted strategy in paleoceanography research. This method is equally applicable to loess research, as evidenced by its successful application in various studies involving CLP sites. Specifically, this approach proves effective when tuning is conducted with one proxy record, while other proxy records reveal cycles distinct from the one used for tuning[52]. The timescale of the Madang profile was established by lithostratigraphic correlation with the Luochuan loess profile based on MS, and the chronology of the latter was constrained by orbital tuning. This approach may lead to circular reasoning in the context of cycle analysis. However, orbital tuning has not resulted in a serious distortion of the actual timescale. The reliability of the orbital-tuned chronology of the Luochuan loess profile has been validated by absolute dating of abundant OSL samples over the last three glacial–interglacial cycles[53].

The FeD/FeT of Madang and the $\delta^{18}O$ of stalagmites in Sanbao Cave exhibit a weak negative correlation ($r_{xy} = −0.21$), and most of the high FeD/FeT values correspond to relatively negative $\delta^{18}O$ values (Fig. 4c). If stalagmite $\delta^{18}O$ represents the EASM intensity and more negative $\delta^{18}O$ indicates a stronger EASM, i.e., increased continental precipitation[2], then the high FeD/FeT at Madang indicates increased summer monsoon precipitation. This aligns with the observation that FeD/FeT enrichment occurred with increasing precipitation when it ranged from 900 mm to 1720 mm, as reported in modern soil studies in South China[44] (Supplementary Fig. 9). However, there are four periods (~90 ka, ~160 ka, ~170 ka and ~248 ka) in which FeD/FeT is not consistent with the $\delta^{18}O$ record. The reasons for this inconsistency and the weak correlation are presumed to be as follows: First, some errors may be introduced into the temporal analysis of samples from Madang during linear interpolation. Second, because of the large spatial and temporal variations in summer monsoon precipitation[22], precipitation in Madang does not strictly increase when the EASM is strong. Third, chemical weathering at Madang has been influenced by changes in microgeomorphology and ecosystems as loess accumulation has occurred. Fourth, iron minerals might have abnormally migrated downwards due to redox reactions, although there is no visible iron mineral concentrated horizontally in the Madang profile. Nevertheless, it is clear that the FeD/FeT ratio at Madang largely reflects the general variability in precipitation.

The FeD/FeT curve of Madang can be well compared with the low latitude summer solar insolation gradient ($r_{xy} = 0.73$), with high FeD/

FeT values corresponding to high insolation differences (Fig. 4d). This suggests that the precession-dominated insolation variation is the main driver of the FeD/FeT variation at Madang. Enhanced insolation will result in increased land–sea thermal differences in East Asia, which reinforces summer monsoons in East Asia, boosts precipitation in Madang, and strengthens chemical weathering, consequently leading to increased FeD/FeT values.

Two records from the monsoon marginal zone show cyclic variations similar to the East Asian speleothem records. One record of EASM precipitation from the western CLP shows a dominant cycle of 23 kyr over the past 260 kyr[14], while another record of wet–dry variations from the Tengger Desert shows a clear 20-kyr cycle over the past ~300 kyr[31]. These are similar to the dominant 18-kyr cycle identified in this study, suggesting that precipitation (wetness) in the broad monsoon zone has varied synchronously at orbital timescales, with high precipitation (wetness) generally correlating with high solar insolation (Fig. 5).

The temperature and precipitation in East Asia over the past 425 kyr were simulated recently[54]. The simulated MAT in East Asia is mainly dominated by the 100-kyr cycle. The simulated MAP shows a distinct 40-kyr cycle in North China, 20- and 40-kyr cycles in the Yangtze River Valley, and a 20-kyr cycle in South China. This simulation suggests that the importance of the 20-kyr cycle increases gradually from north to south in the East Asian monsoon region. This result supports the interpretation that the dominant 100-kyr and 20-kyr cycles in the Yangtze River Valley reflect temperature and precipitation, respectively. Unlike the simulated MAP in the Yangtze River Valley, our record does not show the 40-kyr cycle.

The reconstructed local seawater oxygen isotopes ($\delta^{18}O_{sw}$) changes from borehole U1429 in the East China Sea suggest that there is no significant precessional cycle associated with runoff and precipitation variations in the Yangtze River region[55]. If the FeD/FeT curve from Madang effectively represents the precipitation variations in the middle and lower reaches of the Yangtze River, our results do not support the interpretation of the $\delta^{18}O_{sw}$ data from borehole U1429.

It is generally accepted that there are two climate modes in the EASM region, namely a cold–dry climate during the glacial period and a warm–wet climate during the interglacial period. If the MS and FeD/FeT records in the profile mainly reflect temperature and precipitation, respectively, it means that temperature and precipitation in Madang vary in different cycles at the orbital scale, with four corresponding climate modes: warm–dry, warm–wet, cold–dry and cold–wet. The present climate is warm–dry, the early Holocene climate was warm–wet, the climate during Marine Isotope Stage 2 (MIS 2) was cold–dry, and the MIS 3 climate was cold–wet. According to the predicted insolation variation[56], the climate in Madang is expected to remain dry until AD 5700, after which it will gradually become wet again until AD 31000.

In summary, the Madang profile in this study provides a 350,000-year high-resolution record from the core zone of the EASM. It affords an opportunity to reassess the cyclicity and dynamics of the EASM across a broad region based on the low-latitude loess deposits. Our results suggest that changes in the EASM, as indicated by precipitation in this region, are mainly forced by precession-dominated insolation changes, which addresses the key aspect of the "Chinese 100 kyr problem".

## Methods
### Measurement of MS and FDS
A Bartington MS2 System was used for the MS and FDS measurements. Dried and weighed samples were stored in a nonmagnetic sample box. Low-frequency MS at 470 Hz and high-frequency MS at 4700 Hz were measured. FDS was defined as the difference between the measured values at low and high frequencies[57].

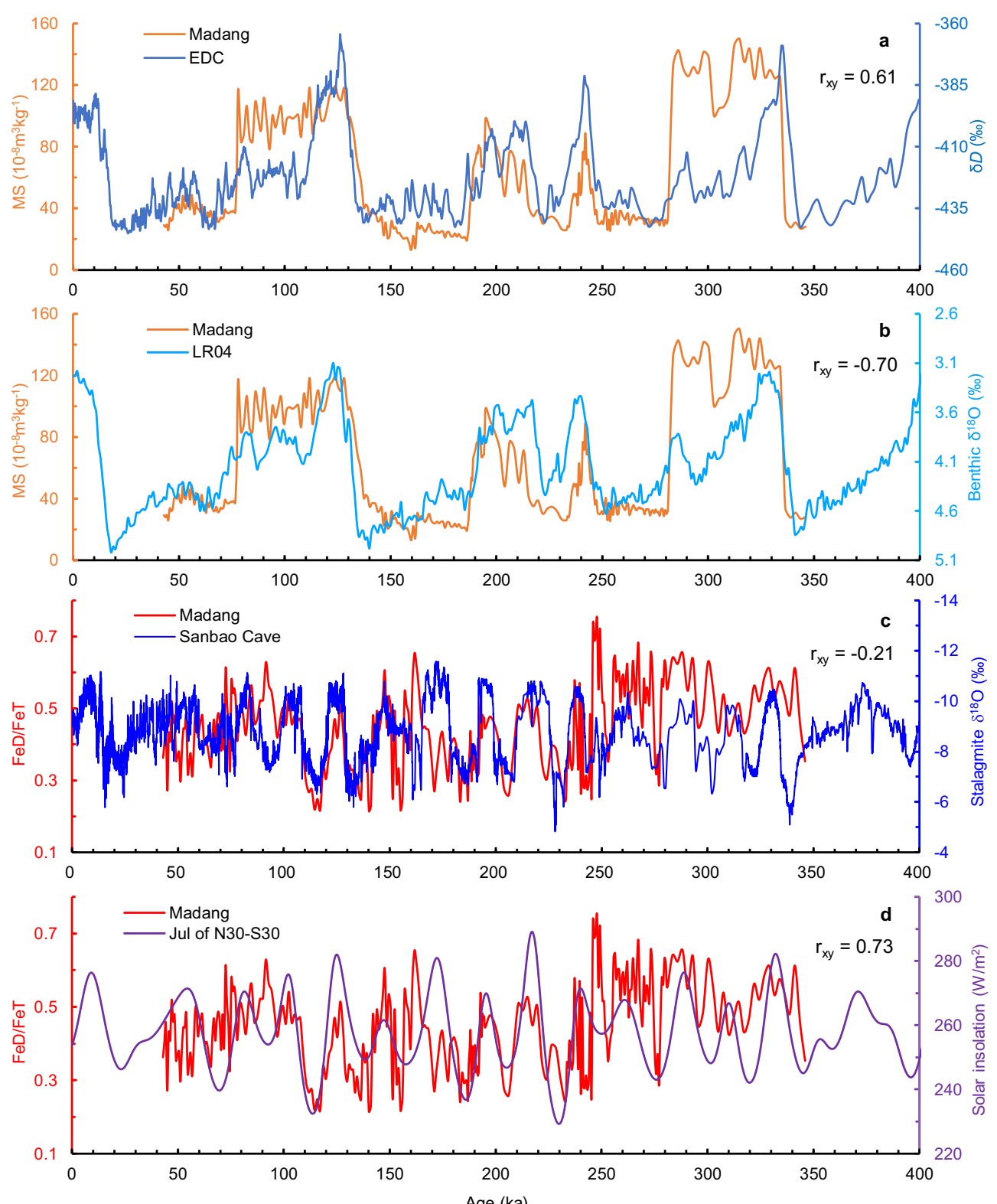

**Fig. 4 | Correlation analysis of the Madang records and other curves. a** Magnetic susceptibility from Madang vs. an Antarctic temperature proxy, $\delta D$[50]. **b** Magnetic susceptibility from Madang vs. benthic $\delta^{18}O$ record from the LR04 stack[51]. **c** FeD/ FeT from Madang vs. stalagmite $\delta^{18}O$ record from Sanbao Cave[18]. **d** FeD/FeT from Madang vs. insolation difference between 30° N and 30° S in July[56].

## Measurement of DCB-extractable iron and total iron

A popular technique used to extract secondary iron oxides is the dithionite−citrate−bicarbonate (DCB) method[58]. Sodium dithionite ($Na_2S_2O_4$) acts as a strong reducing agent, dissolving secondary iron oxides and reducing $Fe^{3+}$ to $Fe^{2+}$, which in turn forms a stable chelate with sodium citrate ($Na_3C_6H_5O_7\cdot2H_2O$); additionally, sodium bicarbonate ($NaHCO_3$) acts as a buffer for the above reaction, keeping the pH of the solution at ~7.3. The original sample was ground in an agate mortar to create a powder sample. Approximately 0.1 g of the sample was weighed (this weight was recorded), and 35 ml of 0.3 N

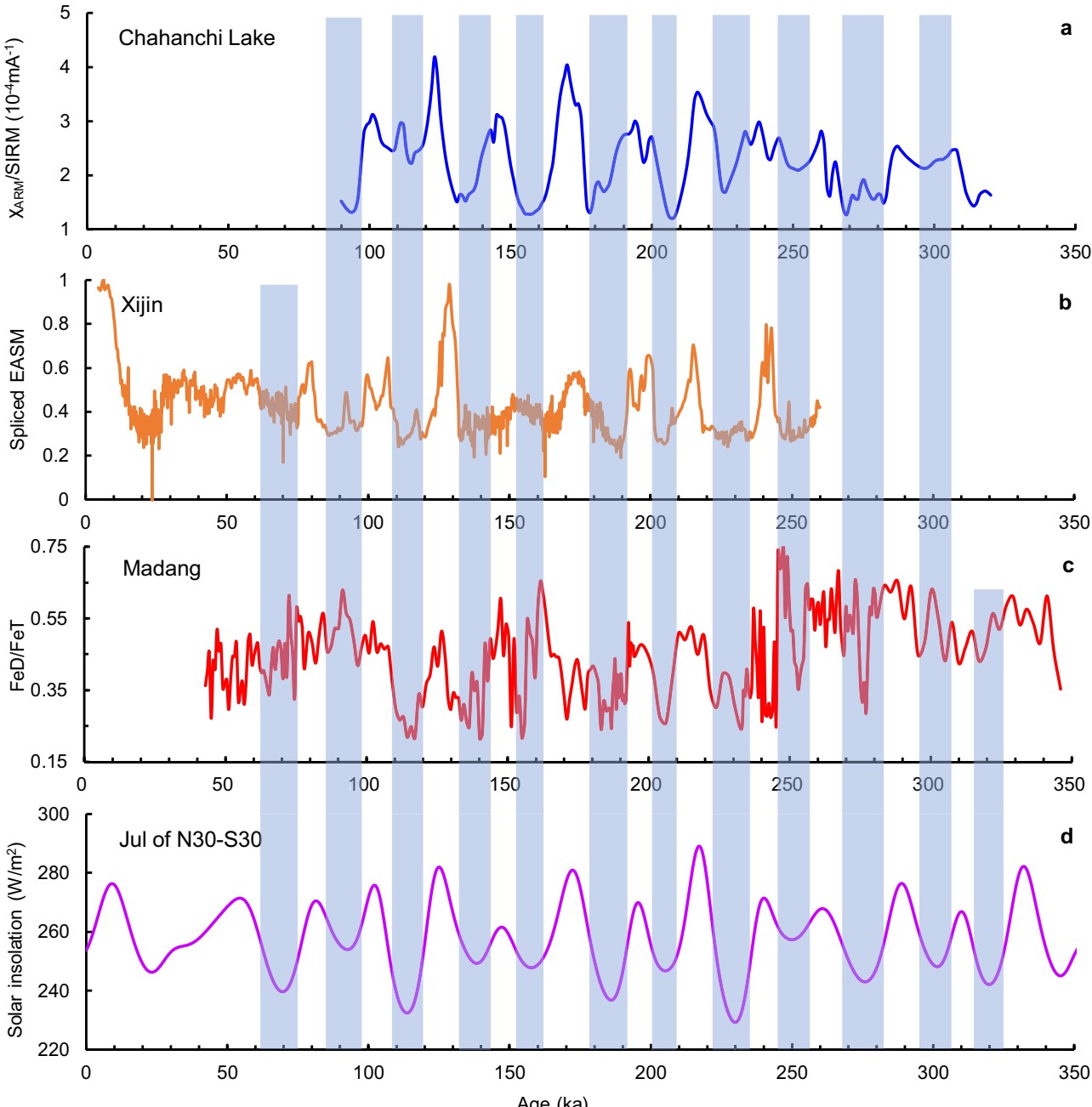

**Fig. 5 | Comparison of the FeD/FeT record from this study with other proxy and forcing time series. a** Wet–dry record from Chahanchi Lake in the Tengger Desert[31]. **b** Spliced East Asian summer monsoon record from Xijin loess drill cores on the western Chinese Loess Plateau[14]. **c** FeD/FeT record from Madang. **d** Insolation difference between 30° N and 30° S in July[56]. The vertical bars indicate intervals of low summer insolation, which generally correspond to periods of low precipitation in both the monsoon core zone (Madang) and the monsoon marginal zone (Chahanchi Lake and Xijin).

sodium citrate solution and 3 ml of 1 N sodium bicarbonate solution were added first, followed by ~2 g of sodium dithionite in excess for the reaction. After sealing, the sample was placed in a shaker at 80 °C and shaken for 90 min until the liquid was gray in color, which indicated that the reaction was complete. Next, the sample was centrifuged at $2795 \times g$ for 15 min, and 5 ml of the supernatant was placed in a small test tube and set aside.

One milliliter of the above liquid was pipetted into a 25 ml colorimetric tube, and 2 drops of 1 N hydrochloric acid and 1.25 ml of 2% (mass%) ascorbic acid solution were added in turn; then, the samples were shaken well. After waiting for 10 min, 2.5 ml of 0.2% O-phenanthroline solution and 4 ml of 25% sodium acetate solution were added, and the samples were shaken well before fixing the

volume. Then, $Fe_2O_3$ solution (100 mg/ml) was used to prepare standard solutions with concentrations of 0, 0.5, 1, 1.5, 2, 2.5 and 3 mg/l. The measured absorbance of the sample was compared with the standard curve obtained from the standard solution to calculate the DCB-extractable iron content. The instrument used was a Shimadzu UV-1800 UV–VIS spectrophotometer with a colorimetric analysis wavelength of 535 nm. Sample measurements were performed at the Ministry of Education Key Laboratory of Surficial Geochemistry, Nanjing University.

Total iron analyses were performed at Nanjing FocuMS Technology Co. Ltd. Approximately 40 mg of powder was mixed with 0.5 ml of 60 wt% $HNO_3$ and 1.0 ml of 40% HF in high-pressure PTFE bombs. These steel-jacketed bombs were then placed in an oven at 195 °C for

3 days to ensure complete digestion. After cooling, the bombs were opened, dried on a hotplate, re-dissolved with 5 ml of 15 wt% $HNO_3$ and 1.0 ml of Rh internal standard, sealed and placed in an oven at 150 °C overnight.

Aliquots of the digestions (dilution factor 500) were introduced into an Agilent Technologies 5110 ICP–OES (Penang, Malaysia) to determine total iron.

## OSL age determination

All OSL samples were pretreated and measured at the Luminescence Dating Laboratory of Nanjing University. Under subdued red light, the bulk materials were initially treated with 10% hydrochloric acid (HCl) and 30% hydrogen peroxide ($H_2O_2$) to remove carbonates and organic material, respectively. The fine grains (4–11 μm) were separated by settling according to Stokes' Law and treated with 30% hydrofluorosilicic acid ($H_2SiF_6$) for 5–7 days to extract the quartz component. The purity of the quartz fractions was verified using the OSL-IR depletion ratio method[59]. Samples with OSL-IR depletion ratios deviating from unity by 10% or more were etched again.

The quartz equivalent dose ($D_e$) was determined based on a standard SAR protocol[60,61], using Risø TL/OSL-DA-20 luminescence readers equipped with blue LEDs (470 nm, ~80 mW cm$^{-2}$) and infrared LEDs (870 nm, ~135 mW cm$^{-2}$) and calibrated $^{90}Sr/^{90}Y$ beta source[62]. A preheat of 260 °C for 10 s and a cut heat of 220 °C were used. The OSL signal was stimulated by blue LEDs at 125 °C for 40 s and detected using a 7.5 mm thick Schott U-340 glass filter.

If two or more aliquots yielded $D_e$ values larger than two times the $D_0$ value of the growth function[63], then the OSL signal of this sample was considered saturated, and therefore, any calculated age was considered the minimum age. The $D_e$ value was given as the weighted arithmetic mean of each set of aliquots and presented alongside the standard error.

The concentrations of uranium, thorium, and potassium were measured using inductively coupled plasma–mass spectrometry (ICP–MS) and atomic emission spectrometry (AES). The water content was obtained by weighing the sample before and after drying, with an uncertainty of ± 10%. For the dose rates of quartz, an alpha efficiency value of 0.04 ± 0.02[64] was adopted. The cosmic ray contribution to the dose rate was estimated as a function of sample elevation and burial depth[65]. Then, all the measured radioactive element concentrations were used to determine the external dose rates of quartz using the conversion factors[66].

## Data availability

The relevant data for the Madang profile are available within the paper and its Supplementary Information file. Source data are provided with this paper.

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

## Acknowledgements

This study was funded by the National Natural Science Foundation of China (Grant Nos. 42021001, 41920104005, 41571188 and 41671191) and the STINT-NSFC mobility Grant (42111530183 and CH2020-8688). Anqi Lyu, Yang Li, Bin Zhang and Chengwei Jin participated in the field investigation.

## Author contributions

Z.H. and X.L. conceived the study. X.Y., Y.W.Z., L.Q. and M.L. performed the experiments. S.Y. improved the OSL chronological experiment. Y.W.Z. and Y.Q.Z. performed the data analysis. H.L. provided financial support and reviewed and edited the manuscript. X.L. and Z.H. wrote and edited the paper with input from all authors.

## Competing interests

The authors declare no competing interests.

## Additional information

Zhiyong Han.

**Peer review information** *Nature Communications* thanks Junsheng Nie
and the other, anonymous, reviewer(s) for their contribution to the peer
review of this work. A peer review file is available.

