## [Peer Review File · Nature Communications]

Loess deposits in the low latitudes of East Asia reveal the ~20-kyr precipitation cycleEditorial Note: Parts of this Peer Review File have been redacted as indicated to remove third-party material where no permission to publish could be obtained.

Reviewer #1 (Remarks to the Author):

The manuscript entitled "Loess deposit reveals variations of 100-kyr cycle of temperature and 20-kyr cycle of precipitation at the low latitude of East Asia" by Li et al. presents a new set of proxy records reconstructed from the loess deposited over the past ~350 ka at Madang, East Asia. In comparison with loess records from the Chinese Loess Plateau, they suggested that the magnetic susceptibility (MS) record of loess varied at 100-kyr and 40-kyr cycles is essentially a proxy of temperature, while the DCB-extractable iron (FeD) varied at 50-kyr, 20-kyr and 10-kyr cycles is actually a proxy of precipitation. On the basis of these interpretations, the authors suggested that the Asian summer monsoon indicated by the precipitation is mainly forced by the summer insolation. Overall, the new data are valuable and the results/interpretations appear to be sound and interesting. I only have a few suggestions/comments for the improvement pending on which I recommend acceptance of this paper.

(1) It might be oversimple to attribute the proxies, FeD, MS, hematite content, and redness, to effects from just temperature and precipitation. In theory, the pedogenesis duration also needs to be accounted for as suggested by Cheng et al. (2022).

(2) The authors suggested that the Madang MS record, varied at 100-kyr and 40-kyr cycles, is basically a proxy of global temperature. This is certainly possible, but the evidence seems tentative, or insufficient.

(3) All the discussions in the manuscript were presented more or less in a qualitative way, for example, lacking assessment of data uncertainties, and missing correlation coefficients for all comparisons, etc.

(4) I am not sure to what degree the circular reasoning could occur. This is largely because all the chronologies involved are essentially from the orbital tuning strategy. I suggest having a short discussion.

(5) Further editing is necessary. For example, the abbreviation DCB needs an elucidation when appears for the first time in the text; and Figs. S8 and S9 seems not be cited anywhere.

Reviewer #2 (Remarks to the Author):

Large attempts had been made to solve the important "Chinese 100-kyr Problem", for example, Cheng et al. made systematic reviews and new attempts to explain and reconcile the related disputes (Cheng et al., 2020 Science China, 2022 The Innovation). The authors aimed at this important issue and provided a loess record from the lower reaches of the Yangtze River spanning past ~350 ka. The magnetic susceptibility (MS) and DCB extractable iron were measured and considered as temperature and precipitation proxies. Spectrum analysis on these records displayed a dominating of 100-kyr and 40-kyr cycles for the MS while 50-kyr, 20-kyr and 10-kyr cycles for the DCB records. They finally draw a conclusion that the temperature and precipitation in the past ~350 ka at the low latitude of East Asia have varied at different cycles, and the change of East Asian summer monsoon indicated by the precipitation in this region is mainly forced by the insolation change due to a variation in the precession. This study will be important if they do provide representative temperature and precipitation proxy and can provide their distinct periodicity evolution pattern. However, the present paper bears some flaws that harm the reliability of the results and the related conclusions. Therefore, the paper needs a major modification to resolve the following problems and concerns.

1. The very basis of this study is the age model. Especially when the authors talk about the 20 - kyr, 10-kyr cycles using spectrum analysis, an accurate age model is indeed needed to convince us of the reliability of the results and relating conclusions. However, the age model of this 15.9 m thick Madang profile is only based on 2 OSL ages on the top part of the section at 2.4 and 3.5 m, in combination with 9 tie points yielding from a visual comparison between the studied MS curve to the MS of Luochuan profile from the Chinese Loess Plateau (Lu et al., 1999). Not to mention whether some hiatus existed in this section, 2 OSL age only on the top part cannot convince me of the subsequent visual comparison and the accuracy of the 9 tie points. For example: the correlation of the first tie point and the following linear interpolation will cause some errors due to different sedimentation rates because the top age of the Madang MS is unknown. The correlation of the fifth tie point is also inappropriate from the shape of the two curves.

2. This age model cannot convince the robustness of spectrum analysis results. Furthermore, for the spectrum result of the MS record (Fig. 3b), the most significant cycles are ~66-kyr and ~33-

kyr instead of 39-kyr and 97-kyr discussed by the authors. The dominating 10-kyr cycle in the DCB Fe record is not significant enough to pass the 95% significant level. For the wavelet results, to keep consistent with the power spectrum result and the fact that the main focus of this study is 100 and 97-kyr cycles, the 100-kyr period should at least be displayed in the wavelet figure (Y-axis), but the current y limit is only ~64 kyr (Fig. 3).

3. Another key conclusion of this study is that the authors proposed the MS and DCB extractable iron as separated temperature and precipitation proxies. For the MS, this conclusion is derived from a visual comparison between the Madang MS curve with the temperature curve of the Antarctic ice core (Fig. 4a). This kind of visual comparison is usually subjective, for example, the reviewer considers that the correlation since 220 ka is ok, but that between 220-320 ka cannot support a real high correlation. In addition, a visual comparison between MS, redness and hematite content are served as supporting evidence. While, hematite generally develops best under warm and dry conditions, forming from ferrihydrite by a dehydration/rearrangement process (Kämpf and Schwertmann, 1983; Maher, 1986) and controlled by multiple factors, such as temperature variations (e.g., Schwertmann, 1971; Torrent et al., 2006; Gao et al., 2018), pedogenesis processes in the drainage area, post-depositional dissolution of detrital magnetic minerals and changes in provenance (Yang & Ding, 2003; Fang et al., 2015). Redness of sediments is considered controlled by the content of fine-grained hematite (Schwertmann et al., 1982; Torrent et al., 1983). Therefore, for such an important conclusion, simple visual comparison with Antarctic ice core records as well as hematite and redness records controlled by multi-factors, should be cautious. More evidence, data or comparison methods are suggested to improve this key conclusion's robustness.

4. In south China, the strong monsoon and potential fluvial process may alter the eolian deposition process, thus leading to some potential hiatus. Further, the potential iron mineral downward migration due to redox reactions may lead to the mismatch or lag between the iron mineral-based proxy and other records in such a humid and wetting region. For example, the highest FeD content is at ~250 ka, but this peak is at top of the L3, which cannot match any proxy (e.g., MS, Sanbao oxygen isotope), but lags behind a strong MS peak at ~240 ka (corresponding to S2 bottom).

5. It is strange that the authors use hematite and redness to reflect temperature but FeD to reflect East Asian Summer Monsoon. Because most FeD in loess is composed of hematite and goethite. Further, redness can mostly reflect the hematite content. Another point is that during the loess stage, the FeD values are slightly higher than those in the paleosols, so can we expect that the overall weathering intensity is strong at loess? If not, how can we believe the short-term variation reflects the weathering intensity or summer monsoon rainfall? In this regard, such inconsistencies indeed question the validity of the FeD as a proxy of iron mineral weathering in this study. Rather, the FeD may also reflect the overall provenance effect on iron mineral content at loess and paleosols horizons. Because the authors used FeD rather than the FeD/FeT ratio to reflect summer monsoon precipitation, which means that the potential provenance change in iron minerals cannot be excluded. Such provenance effect may exhibit a profound control on changes in the orbital variations.

RESPONSE TO REVIEWERS

We thank the reviewers for providing valuable comments and insightful suggestions on our manuscript. These comments and suggestions have been extremely helpful in improving the quality of our work. After careful consideration of the comments, we have fully revised our manuscript to address all of the reviewers' concerns. Our detailed responses are set out below (highlighted in blue). The main changes have been made in the following areas:

- (1) Four additional OSL samples were measured, and new OSL ages were used to prove the validity of the first tie point (56 ka) of the timescale;
- (2) Total iron (FeT) values were measured for 291 samples, and the FeD proxy indicator is replaced by the FeD/FeT ratio in order to eliminate the provenance effect;
- (3) The fifth tie point of the age controls was adjusted from the depth of 820 cm to 860 cm, as suggested. Similarly, the third tie point was moved from the depth of 460 cm to 445 cm. As a result, the timescale of the Madang profile was revised accordingly;
- (4) When comparing time series of different proxy indicators, the cross-correlation coefficient (r_{xy}) was calculated using a program specifically designed for unevenly spaced time series;
- (5) Uncertainty for each proxy indicator was estimated by replication analysis or comparison with standard (known) samples;
- (6) The Vostok temperature curve was replaced by the EDC curve because the timescale of the EDC core was well constrained.

Reviewer #1 (Remarks to the Author):

The manuscript entitled "Loess deposit reveals variations of 100-kyr cycle of temperature and 20-kyr cycle of precipitation at the low latitude of East Asia" by Li et al. presents a new set of proxy records reconstructed from the loess deposited over the past ~350 ka at Madang, East Asia. In comparison with loess records from the Chinese Loess Plateau, they suggested that the magnetic susceptibility (MS) record of loess varied at 100-kyr and 40-kyr cycles is essentially a proxy of temperature, while the DCB-extractable iron (FeD) varied at 50-kyr, 20-kyr and 10-kyr cycles is actually a proxy of precipitation. On the basis of these interpretations, the authors suggested that the Asian summer monsoon indicated by the precipitation is mainly forced by the summer insolation. Overall, the new data are valuable and the results/interpretations

appear to be sound and interesting. I only have a few suggestions/comments for the improvement pending on which I recommend acceptance of this paper.

(1) It might be oversimple to attribute the proxies, FeD, MS, hematite content, and redness, to effects from just temperature and precipitation. In theory, the pedogenesis duration also needs to be accounted for as suggested by Cheng et al. (2022).

Response: We appreciate the reviewer's helpful comments. When we state that an indicator can be interpreted as a proxy for temperature (precipitation), we do not mean that the indicator is a pure proxy for temperature (precipitation). A more precise expression is that the indicator can be interpreted as a temperature-dominated (precipitation-dominated) proxy. Regardless of whether they are biological or geochemical indicators, it is important to acknowledge that their changes are influenced by a variety of factors from different perspectives. Taking climate as an example, the change in biological or geochemical indicators is mainly driven by the seasonal combination of water and heat. It is actually impossible to extract an indicator as a pure proxy for temperature (precipitation). Ideally, an indicator can only be interpreted as a temperature-dominated (precipitation-dominated) proxy. To avoid misunderstandings, we have revised our expression in the text by adding "-dominated". The pedogenesis duration suggested by Cheng et al. (2022) was added as an effect; and much analyses were added to the revised manuscript.

(2) The authors suggested that the Madang MS record, varied at 100-kyr and 40-kyr cycles, is basically a proxy of global temperature. This is certainly possible, but the evidence seems tentative, or insufficient.

Response: In principle, pedogenesis is generally considered to be the mechanism that enhances magnetic susceptibility (MS). However, pedogenesis is largely associated with biological activity which is controlled by temperature, precipitation, their seasonal combination, and parent materials, etc. Due to the spatially synchronous change of precipitation and temperature in East Asia, it is difficult to determine which factor is the main driving force by comparing the MS of modern surface samples only. Therefore, we cannot directly prove that the Madang MS is a proxy for temperature in terms of mechanism. Nevertheless, since the change in loess MS at low latitude (Madang) is similar to that at mid-latitude (Chinese Loess Plateau), this implies that loess MS is a proxy for a climatic factor that can show a consistent variation over a large area (from mid-latitude to low latitude). In this sense, the Madang MS is more likely to be a proxy for temperature than for precipitation, because temperature presents a large space

consistent during the glacial-interglacial alternations in Northern Hemisphere, whereas precipitation often shows a spatial discrepancy. Assuming that the Madang MS is a temperature-dominated proxy, its change must be generally similar to the known temperature record. Since the Madang MS shows a very similar trend of change to global temperature, it is reasonable to deduce that the Madang MS is a temperature-dominated proxy.

(3) All the discussions in the manuscript were presented more or less in a qualitative way, for example, lacking assessment of data uncertainties, and missing correlation coefficients for all comparisons, etc.

Response: We thank the reviewer for bringing this issue to our attention. To address the concern, we have performed uncertainty assessments for all data and compared the results with the variation amplitude of each indicator in the Madang profile (Table S3). It is evident that the data uncertainty is significantly smaller than the variation amplitude, demonstrating that the observed indicator variation in the profile is true to reveal the past climate change. We also performed correlation analyses between the two time series. Using a computational program designed for unevenly spaced time series (Polanco-Martinez, et al., 2019), we estimated the cross-correlation coefficient (r_{xy}) between the different time series. The correlation coefficients are presented in Figure 4 and discussed in the text.

Tab. S3 Estimated uncertainty for each proxy parameter measured in this study

Parameter	Estimation method	Aliquots	Samples	Relative error	Standard deviation	Error	Variation in Madang profile
Mean grain size (Mz)	Replication analysis	24			0.4 μm		10.6-13.5 μm
Magnetic susceptibility (MS)	Replication analysis	20			$0.5 \times 10^{-8} \text{m}^3 \text{kg}^{-1}$		$13-150 \times 10^{-8} \text{m}^3 \text{kg}^{-1}$
DCB-extractable iron (FeD)	Replication analysis	21			0.02%		1.16-4.63%
Total iron (FeT)	Standard sample			<2%			4.14-6.38%
Hematite (Hm)	Known sample		7			0.03%	0.19-0.88%
Redness (a^*)	Replication	60			0.12		6.82-10.77

(4) I am not sure to what degree the circular reasoning could occur. This is largely because all the chronologies involved are essentially from the orbital tuning strategy. I suggest having a short discussion.

Response: Thanks for the useful suggestion. A paragraph has been added to the Discussion section to address this issue, namely “The timescale of the Madang profile was established by magnetic susceptibility stratigraphic correlation with the Luochuan loess profile, the chronology of which was constrained by orbital tuning. This approach may lead to circular reasoning for the cycle analysis. However, the orbital tuning did not result in a significant distortion of the actual timescale. The reliability of the orbital-tuned chronology of the Luochuan loess profile has been validated by the absolute dating of densely sampled OSL samples over the last three glacial-interglacial cycles (Zhang et al., 2022)”. (See Fig. R1 below).

Fig. R1 Bacon age-depth model for the Luochuan section of the Chinese Loess Plateau (Zhang et al., 2022).

(5) Further editing is necessary. For example, the abbreviation DCB needs an

elucidation when appears for the first time in the text; and Figs. S8 and S9 seems not be cited anywhere.

Response: Abbreviations were expanded at first use. The citations were inserted into text.

Reviewer #2 (Remarks to the Author):

Large attempts had been made to solve the important “Chinese 100-kyr Problem”, for example, Cheng et al. made systematic reviews and new attempts to explain and reconcile the related disputes (Cheng et al., 2020 Science China, 2022 The Innovation). The authors aimed at this important issue and provided a loess record from the lower reaches of the Yangtze River spanning past ~350 ka. The magnetic susceptibility (MS) and DCB extractable iron were measured and considered as temperature and precipitation proxies. Spectrum analysis on these records displayed a dominating of 100-kyr and 40-kyr cycles for the MS while 50-kyr, 20-kyr and 10-kyr cycles for the DCB records. They finally draw a conclusion that the temperature and precipitation in the past ~350 ka at the low latitude of East Asia have varied at different cycles, and the change of East Asian summer monsoon indicated by the precipitation in this region is mainly forced by the insolation change due to a variation in the precession. This study will be important if they do provide representative temperature and precipitation proxy and can provide their distinct periodicity evolution pattern. However, the present paper bears some flaws that harm the reliability of the results and the related conclusions. Therefore, the paper needs a major modification to resolve the following problems and concerns.

1. The very basis of this study is the age model. Especially when the authors talk about the 20 -kyr, 10-kyr cycles using spectrum analysis, an accurate age model is indeed needed to convince us of the reliability of the results and relating conclusions. However, the age model of this 15.9 m thick Madang profile is only based on 2 OSL ages on the top part of the section at 2.4 and 3.5 m, in combination with 9 tie points yielding from a visual comparison between the studied MS curve to the MS of Luochuan profile from the Chinese Loess Plateau (Lu et al., 1999). Not to mention whether some hiatus existed in this section, 2 OSL age only on the top part cannot convince me of the subsequent visual comparison and the accuracy of the 9 tie points. For example: the correlation of the first tie point and the following linear interpolation will cause some errors due to different sedimentation rates because the top age of the

Madang MS is unknown. The correlation of the fifth tie point is also inappropriate from the shape of the two curves.

Response: We appreciate this comment and agree with the reviewer's emphasis on ensuring the accuracy of the chronology. In fact, the age model of the Madang profile was established by linear extrapolation and interpolation of 9 tie points determined by MS stratigraphic correlation. The two OSL ages were only used to check the validity of the stratigraphic correlation in the original manuscript. We agree with the reviewer that the first tie point cannot be well verified by the two OSL ages. Therefore, we performed age measurements on four additional OSL samples. The results (Tab.S4, in red) show that these four OSL ages are consistent with the stratigraphic sequence. The first tie point (56 ka) falls exactly in the period between 39.6 ± 2.7 ka and 66.8 ± 4.8 ka (as shown in the updated Fig. 2). The new OSL ages provide further evidence for the reliability of the first tie point.

Tab. S4 OSL dating results of quartz of Madang profile

Sample	Depth (cm)	Grain size (μm)	Water content (%)	U (ppm)	Th (ppm)	K (%)	De (Gy)	Dose rate (Gy/ka)	Age (ka)	Aliquot number
A8-40	40	4-11	17.1	114.0 ± 2.3	114.0 ± 2.3	114.0 ± 2.3	114.0 ± 2.3	4.03 ± 0.28	30.1 ± 2.2	6
A17-85	85	4-11	17.1	149.6 ± 1.8	149.6 ± 1.8	149.6 ± 1.8	149.6 ± 1.8	4.12 ± 0.28	39.6 ± 2.7	6
MD-P-0.6	155	4-11	13.2	3.06 ± 0.15	13.42 ± 0.67	1.75 ± 0.03	246.7 ± 4.2	3.69 ± 0.26	66.8 ± 4.8	6
MD-P-1.2	215	4-11	15.4	3.07 ± 0.15	13.66 ± 0.68	1.71 ± 0.03	262.6 ± 9.3	3.58 ± 0.25	73.3 ± 5.8	6
MD-P-1.5	245	4-11	17.4	3.21 ± 0.12	13.90 ± 0.38	1.71 ± 0.05	271.5 ± 5.2	3.56 ± 0.25	76.3 ± 5.5	6
MD-P-4.5	350	4-11	17.6	2.44 ± 0.10	15.40 ± 0.42	1.96 ± 0.06	312.8 ± 6.7	3.65 ± 0.25	85.6 ± 6.0	6

Updated Fig. 2 showing correlation of magnetic susceptibility curves of the Madang profile at low latitude and the Luochuan profile at middle latitude (Lu et al., 1999).

Four data in the red circle are newly measured OSL ages.

Based on the characteristics of two MS curves, we made adjustments to the third and fifth tie points. As suggested, the fifth tie point was shifted from the depth of 820 cm to 860 cm. Similarly, the third tie point was shifted from the depth of 460 cm to 445 cm. These modifications resulted in the generation of a new timescale, which was then used for further analysis (Fig. R2). The cross-correlation coefficient (r_{xy}) between the MS time series of the Madang and Luochuan profiles was calculated to be 0.88, indicating the reliability of the Madang timescale (Fig. R3).

Fig. R2 Old and new timescales of the Madang profile

Fig. R3 Comparison of Madang and Luochuan MS curves. The strong correlation ($r_{xy} = 0.88$) demonstrates that the timescale of Madang, generated using nine tie points, is reliable.

Response: The overall similarity in the stratigraphic and proxy records across numerous profiles suggests that, generally, loess accumulation is likely continuous on orbital timescales throughout the main body of the Chinese Loess Plateau (Zhang et al., 2022; Lu et al., 2022). This suggestion is further supported by densely-sampled OSL dating, which reveals that loess accumulation at Luochuan was continuous over the last three glacial–interglacial cycles on orbital timescales (Fig. R1). Several factors lead us to believe that the Madang profile is continuous on orbital timescales. These include the overall similarity in MS stratigraphy (updated Fig. 2), the absence of erosion surfaces,

and the lack of abnormal sedimentation rates as illustrated in the age-depth plot (Fig. R2).

The records (MS, FeD/FeT, and Hm) analyzed with the new timescale reveal clearer orbital cycles (refer to updated Fig. 3 and Fig. S8). Specifically, both MS and Hm curves exhibit a dominant cycle of 101-kyr, while the FeD/FeT curve displays a distinct cycle of 18-kyr. It is evident that the former can be interpreted as a temperature-dominated cycle, whereas the latter can be attributed to a precession-dominated cycle.

Updated Fig. 3 Wavelet and spectral analysis of MS (a, b) and FeD/FeT (c, d) of the Madang profile

Updated Fig. S8 Wavelet and spectral analysis of the temperature proxy (δD) of the Antarctic ice core (a, b) and the hematite (Hm) content of the Madang profile(c, d)

2. This age model cannot convince the robustness of spectrum analysis results. Furthermore, for the spectrum result of the MS record (Fig. 3b), the most significant cycles are ~ 66 -kyr and ~ 33 -kyr instead of 39-kyr and 97-kyr discussed by the authors. The dominating 10-kyr cycle in the DCB Fe record is not significant enough to pass the 95% significant level. For the wavelet results, to keep consistent with the power spectrum result and the fact that the main focus of this study is 100 and 97-kyr cycles, the 100-kyr period should at least be displayed in the wavelet figure (Y-axis), but the current y limit is only ~ 64 kyr (Fig. 3).

Response: We agree that the extracted cycles in the original manuscript were not convincing, mainly due to the inappropriate selection of the third and fifth tie points. However, by adjusting these tie points, we have derived a new sequence that produces cycles that clearly exceed the 95% significance level and can be readily correlated with orbital cycles (updated Fig.3).

We appreciate the reviewer's suggestion and have rescaled the y-axis of the wavelet plot to better show the 100 kyr period.

3. Another key conclusion of this study is that the authors proposed the MS and DCB extractable iron as separated temperature and precipitation proxies. For the MS, this conclusion is derived from a visual comparison between the Madang MS curve with the temperature curve of the Antarctic ice core (Fig. 4a). This kind of visual comparison is usually subjective, for example, the reviewer considers that the correlation since 220 ka is ok, but that between 220-320 ka cannot support a real high correlation. In addition, a visual comparison between MS, redness and hematite content are served as supporting evidence. While, hematite generally develops best under warm and dry conditions, forming from ferrihydrite by a dehydrationrearrangement process (Kämpf and Schwertmann, 1983; Maher, 1986) and controlled by multiple factors, such as temperature variations (e.g., Schwertmann, 1971; Torrent et al., 2006; Gao et al., 2018), pedogenesis processes in the drainage area, post-depositional dissolution of detrital magnetic minerals and changes in provenance (Yang & Ding, 2003; Fang et al., 2015). Redness of sediments is considered controlled by the content of fine - grained hematite (Schwertmann et al., 1982; Torrent et al., 1983). Therefore, for such an important conclusion, simple visual comparison with Antarctic ice core records as well as hematite and redness records controlled by multi-factors, should be cautious. More evidence, data or comparison methods are suggested to improve this key conclusion's robustness.

Response: We thank the reviewer for this constructive suggestion. We agree that visual comparison alone can be subjective. It is easy to quantify the potential association between two evenly spaced climate time series, as with Pearson's and Spearman's correlation analyses, etc. However, these methods are not available when the time series are unevenly spaced, especially when the two time series under analysis are not sampled at identical points in time. Fortunately, Polanco-Martinez et al. (2019) have developed a computational program called BINCOR to quantify the correlation between two unevenly spaced time series. It is based on a novel method proposed by Mudelsee (2010) for estimating the correlation between two climate time series with different timescales. Therefore, in revised manuscript we introduced the BINCOR program for correlation analysis in order to make a rigorous comparison. The cross-correlation coefficient (r_{xy}) was obtained and then added to Fig. 4 and discussed in the text. There is a relatively strong correlation between the Madang MS and temperature proxies ($r_{xy} = 0.61$), and between the Madang FeD/FeT and insolation proxies ($r_{xy} = 0.73$). We also calculated the cross-correlation coefficient between MS, Hm and a^* . The results show r_{xy} is 0.54 between Madang MS and hematite content, 0.90 between Madang MS and redness. The

relatively high correlation coefficients show the compared indicators have close relations, which can support our interpretation.

Updated Fig. 4 Correlation of the Madang's records with other paleoclimate curves.

The Vostok record is replaced by the EDC record and the cross-correlation coefficients are also given.

As the reviewer pointed out, we have also noticed that the comparison between the Madang MS curve and the Antarctic ice core temperature curve during 220-320 ka is unsatisfactory. This discrepancy probably due to the improperly constrained timescale of the Vostok core. In reviewing other Antarctic ice cores, we have found that the timescale of the EDC core is well constrained (Fig. R4). In addition, temperature proxies, specifically the δD (deuterium/hydrogen ratio) in the ice, have been measured for the EDC core by the EPICA community members (2004). Therefore, we have replaced the Vostok data with the EDC core data to improve the comparison in the revised manuscript.

Fig. R4 Comparison between δD from EPICA Dome C and δD from Vostok (EPICA community members, 2004)

We agree with the comment that geochemical, geophysical, and biological indicators are influenced by multiple factors. Currently, it is actually impossible to extract an indicator that purely represents temperature or precipitation. Ideally, an indicator can only be interpreted as a temperature-dominated or precipitation-dominated proxy. To avoid misunderstandings, we have revised our expression in the text by adding "-dominated", with much analyses and interpretations in the revised manuscript.

4. In south China, the strong monsoon and potential fluvial process may alter the eolian deposition process, thus leading to some potential hiatus. Further, the potential iron mineral downward migration due to redox reactions may lead to the mismatch or lag between the iron mineral-based proxy and other records in such a humid and wetting region. For example, the highest Fed content is at ~ 250 ka, but this peak is at top of the L3, which cannot match any proxy (e.g, MS, Sanbao oxygen isotope), but lags behind a strong MS peak at ~ 240 ka (corresponding to S2 bottom).

Response: We have examined the Madang loess-paleosol profile for possible hiatuses. Since there is no erosion surface, no abrupt change in sedimentary facies, and no abrupt change in MS and grain size curves, we suggest that the Madang loess is continuous at orbital timescales. We agree that the possible vertical migration of iron minerals may lead to a mismatching. We have studied many other loess profiles in the lower reaches of the Yangtze River and usually found iron minerals concentrated in some horizons (Fig. R5). The MS curve of these profiles is not similar to that of the Luochuan profile, and the vertical migration of iron minerals is thought to be one of the causes. Fortunately, the Madang profile has no obvious concentration of iron minerals.

Fig. R5 Horizontally concentrated iron minerals in Xiashu loess near Yangzhou in the lower reaches of the Yangtze River

It is not clear that the highest FeD/FeT content at ~248 ka does not coincide with a $\delta^{18}\text{O}$ peak of the stalagmites. As we mentioned in the text, there are four periods (~90 ka, ~160 ka, ~170 ka, and ~248 ka) where the FeD/FeT is inconsistent with the $\delta^{18}\text{O}$. We have proposed three causes for this inconsistency and the weak correlation between Madang MS and Sanbao $\delta^{18}\text{O}$. First, the temporal analysis of the Madang samples may introduce some errors during linear interpolation. Second, because of the large spatial and temporal variations in summer monsoon precipitation, precipitation at Madang does not strictly increase when the EASM is strong. Third, chemical weathering at Madang has been influenced by changes in microgeomorphology and ecosystems as the loess accumulation has occurred. Now the possible vertical migration of iron minerals was proposed as the fourth cause. The following sentence was added to the text, “Fourth, iron minerals might have abnormally migrated downward due to redox reactions, although there is no visible iron mineral concentrated horizontally in the Madang profile.”

5. It is strange that the authors use hematite and redness to reflect temperature but FeD to reflect East Asian Summer Monsoon. Because most FeD in loess is composed

of hematite and goethite. Further, redness can mostly reflect the hematite content. Another point is that during the loess stage, the FeD values are slightly higher than those in the paleosols, so can we expect that the overall weathering intensity is strong at loess? If not, how can we believe the short-term variation reflects the weathering intensity or summer monsoon rainfall? In this regard, such inconsistencies indeed question the validity of the FeD as a proxy of iron mineral weathering in this study. Rather, the FeD may also reflect the overall provenance effect on iron mineral content at loess and paleosols horizons. Because the authors used FeD rather than the FeD/FeT ratio to reflect summer monsoon precipitation, which means that the potential provenance change in iron minerals cannot be excluded. Such provenance effect may exhibit a profound control on changes in the orbital variations.

Response: We thank the reviewer for raising the potential differences between FeD and FeD/FeT. We agree that the potential provenance change of iron minerals cannot be excluded by using FeD. To eliminate this provenance effect, we re-measured the FeT values of all samples and used the FeD/FeT ratio instead of FeD as the precipitation dominated indicator for analysis. The data show that the FeD/FeT records overall are similar to the FeD records (Fig. R6).

Fig. R6 Comparison between FeD and FeD/FeT

The ratio of FeD/FeT is one of the widely used indicators to evaluate the intensity of soil chemical weathering. It is considered a measure of the quantity of iron liberated from iron-bearing silicate minerals relative to the total iron available (Guo et al., 1998, 2000). The effects of precipitation and temperature on chemical weathering rates must be considered simultaneously. A study in South China revealed a relatively strong positive correlation between FeD/FeT and MAP and no correlation between FeD/FeT and MAT (Fig. S10; Fig. S11; Long et al., 2016), it seems that FeD/FeT is a precipitation-dominated proxy indicator in low-altitude areas of South China.

Fig. S10 Dependence of the ratio of FeD/FeT in modern soil on the mean annual precipitation (MAP) (Long et al., 2016). The FeD/FeT generally increases with MAP up to ~1720 mm and decreases with more MAP (left). A linear correlation is seen for MAP less than 1720 mm (right).

Fig. S11 Relationship of FeD/FeT in modern soil with the mean annual temperature (MAT) (Long et al., 2016). It is clear that the FeD/FeT is independent of MAT

Our recent study shows that dithionite-citrate-bicarbonate (DCB) dissolution of iron oxides in loess-paleosol samples is highly grain size dependent (Fig. R7), and that DCB treatment does not selectively dissolve specific iron oxide mineral species (Yang et al., 2022). FeD is not related to hematite and goethite as was once thought. In fact, FeD is related to fine-grained iron minerals (Fig. R8), which were supposed to be mainly secondary iron minerals formed by oxidation of the iron ions released from aluminosilicates. To clarify the relationships between modern climatic variables and pedogenic hematite, as well as pedogenic ferrimagnetic minerals, Gao et al. (2018) performed a comprehensive investigation of the magnetic properties and statistical analysis of a suite of clay and silt fractions of modern soil samples from 179 sites across the Chinese Loess Plateau and adjacent regions. The results show that the formation of

ferrimagnetic minerals mainly depends on mean annual precipitation (MAP). However, the hematite formation is preferentially dependent on mean annual temperature (MAT). High ambient temperature favors the production of hematite, probably through promoting the transformation of maghemite to hematite, as well as directly from ferrihydrite to hematite (Gao et al., 2018). Therefore, when FeD (FeD/FeT) is considered as a precipitation-dominated proxy in low-altitude areas, it is not contradictory that redness (hematite) can be considered as a temperature-dominated proxy.

Fig. R7. Relative changes in different grain size fractions after DCB treatment to those before DCB treatment (Yang et al., 2022).

Fig. R8 Three iron oxide components in the loess-paleosol LC profile classified according to the response of the magnetic susceptibility to DCB dissolution (Yang et al., 2022).

It is true that the FeD/FeT values can be slightly higher than those in the paleosols in Madang profile, for example, FeD/FeT at ~145 ka (loess) is higher than that at ~100 ka (paleosol). But we cannot conclude that the overall weathering intensity is strong at loess. The weathering intensity is usually determined by the combination of temperature and precipitation. A high rainfall indicated by FeD/FeT cannot solely lead to a strong weathering. In this sense, the weathering cannot be used as a proxy of temperature or rainfall in most cases. Although MS and FeD (FeD/FeT) have been roughly treated as weathering proxies. In fact, they have different enhancement mechanisms. Generally, MS reflects pedogenesis intensity determined by the biological activity, FeD/FeT reflects chemical reaction intensity determined by water activity.

References

- Cheng, H. et al. Milankovitch theory and monsoon. *The Innovation* 3, 100338 (2022).
<https://doi.org/10.1016/j.xinn.2022.100338>
- EPICA community members. Eight glacial cycles from an Antarctic ice core. *Nature* 429, 623–628 (2004). <https://doi.org/10.1038/nature02599>
- Gao, X., Hao, Q., Wang, L., Oldfield, F., Bloemendal, J., Deng, C., Song, Y., Ge, J., Wu, H., Xu, B., Li, F., Han, L., Fu, Y., Guo, Z. The different climatic response of pedogenic hematite and ferrimagnetic minerals: Evidence from particle-sized modern soils over the Chinese Loess Plateau, *Quat. Sci. Rev.*, 179, 69-86 (2018).
<https://doi.org/10.1016/j.quascirev.2017.11.011>
- Guo, Z., Liu, T., Fedoroff, N., Wei, L., Ding, Z., Wu, N., Lu, H., Jiang, W., An, Z. Climate extremes in Loess of China coupled with the strength of deep-water formation in the North Atlantic. *Glob. Planet. Change* 18, 113–128 (1998).
[https://doi.org/10.1016/S0921-8181\(98\)00010-1](https://doi.org/10.1016/S0921-8181(98)00010-1)
- Guo, Z., Biscaye, P., Wei, L., Chen, X., Peng, S., Liu, T. Summer monsoon variations over the last 1.2 Ma from the weathering of loess-soil sequences in China. *Geophys. Res. Lett.* 27, 1751–1754 (2000).
<https://doi.org/10.1029/1999GL008419>
- Lu, H., Liu, X., Zhang, F., An, Z., Dodson, J. Astronomical calibration of loess–paleosol deposits at Luochuan, central Chinese Loess Plateau, *Palaeogeogr. Palaeoclimatol. Palaeoecol.* 154, 237–246 (1999).
[https://doi.org/10.1016/s0031-0182\(99\)00113-3](https://doi.org/10.1016/s0031-0182(99)00113-3)
- Lu, H., Wang, X., Wang, Y., Zhang, X., Yi, S., Wang, X., Stevens T., Kurbanov, R., Markovic, S.B. Chinese Loess and the Asian Monsoon: What We Know and

- What Remains Unknown, *Quaternary International*, 620, 85-97 (2022). <https://doi.org/10.1016/j.quaint.2021.04.027>
- Long, X., Ji J., Barron, V., Torrent, J. Climatic thresholds for pedogenic iron oxides under aerobic conditions: Processes and their significance in paleoclimate reconstruction. *Quat. Sci. Rev.* 150, 264–277 (2016). <https://doi.org/10.1016/j.quascirev.2016.08.031>
- Mudelsee, M. *Climate Time Series Analysis: Classical Statistical and Bootstrap Methods*. Springer-Verlag, ISBN 9048194814 (2010).
- Polanco-Martinez, J. M., Medina-Elizalde, M. A., Sanchez Goni, M, F., Mudelsee, M. BINCOR: An R package for estimating the correlation between two unevenly spaced time series. *The R Journal* 11, 170–184 (2019). <https://doi.org/10.32614/RJ-2019-035>
- Yang, Q., Li, X., Han, Z., Wang X., Zhao, W., Yi, S., Lu, H. DCB dissolution of iron oxides in aeolian dust deposits controlled by particle size rather than mineral species. *Sci. Rep.* 12, 2786 (2022). <https://doi.org/10.1038/s41598-022-06734-2>
- Zhang J.J., Hao Q.Z., Li S.H., An absolutely dated record of climate change over the last three glacial–interglacial cycles from Chinese loess deposits. *Geology*, 50: 1116–1120 (2022). <https://doi.org/10.1130/G50125.1>

Reviewer #1 (Remarks to the Author):

The authors have satisfactorily addressed my concerns and I do not have further comments

Reviewer #2 (Remarks to the Author):

As I suggested in the last version, it would be important in solving the "Chinese 100-kyr Problem", if the authors indeed present high-resolution independent temperature and precipitation records with accurate chronology. However, the current version cannot demonstrate such progress. The authors have tried to solve the concerns I proposed, for example, they added some new OSL dating points and re-correlated the original dating results with the Luochuan magnetic susceptibility (MS) records. However, the key of this study is how to demonstrate that the MS and DCB-extractable iron to total iron are independent temperature and precipitation proxies, respectively.

The main criticism is that the authors used a variety of proxies related to iron minerals to unmix the temperature and precipitation signals in an aeolian sequence. However, the proxy indication of iron minerals in the CLP and Madang may exhibit different controls, which may induce an unclear proxy indication. For example, magnetic susceptibility (MS) is used to reflect summer monsoon intensity in the CLP, and the MS curve can be well correlated throughout the CLP, thus providing a powerful tool for age control. The present study also finds that the Madang MS curve can match that in the CLP well, but the authors think that MS should reflect a dominant temperature response because in Madang there is abundant precipitation and pedogenesis is controlled by a major temperature effect. Such different proxy indications in different regions bring about difficulties to directly separating temperature and precipitation controls due to so close link between them in East Asia monsoon region. Moreover, a global temperature similar pattern does not mean direct temperature control. Given this, a direct temperature reconstruction based on biochemical proxy (e.g., GDGTs) may solve this problem. In addition, I am strange that frequency-dependent magnetic susceptibility Xfd was not used in this study, which exhibits a clearer pedogenesis signal than MS.

What's more, by comparing the temperature of Madang MS with that of the Antarctic ice core, the authors believe that MS is mainly controlled by temperature change. Obviously, such logic is problematic. A large number of studies have indicated that MS is controlled by the combined effects of both temperature and rainfall changes, so it is not surprising that there is a positive correlation with temperature (the R2 value between them in this study is less than 0.4, this cannot be considered as a high correlation). The author does not rule out the influence of rainfall on MS here. If correlation analysis is conducted between MS and local rainfall, I think there may also be a positive correlation. In addition, the cross-correlation coefficient between MS and Hm is meaningless. In the Loess Plateau, MS also has a very good positive correlation with redness and hematite content, and even in the lake basin sediments of the NE Tibetan Plateau, such a high positive correlation exists, because the higher the redness, the higher the hematite or magnetic hematite content in the sedimentary records, and the corresponding MS value is higher. How do the authors rule out the effect of temperature on the FeD/FeT indicator? In fact, the occurrence of unsynchronized changes in rain and heat at the orbital scale is very small in Madang region, which may not be consistent with the current meteorological record. Therefore, is it possible for the author to provide high-resolution palypollen records or other proxies that can directly reflect changes in temperature or rainfall to further support the views of this paper?

Other concerns

1. For the introduction part, the authors did not describe the latest research progress well, and did not systematically summarize the main problems at present, and only cited the MS data of the Loess Plateau to elaborate, which is obviously one-sided.

For example, Guo et al.(2022) have suggested that the MS records in the CLP has a significant precession cycle, at least in the interglacial periods. The followings are some related paper.

Bao, R., Sheng, X., Meng, X., Li, T., Li, C., Shen, H., Da, J., Ji, J., and Chen, J., 2022, 100 k.y. pacing of the East Asian summer monsoon over the past five glacial cycles inferred from land snails: *Geology*, v. 51, no. 2, p. 179-183.

Guo, B., Nie, J., Stevens, T., Buylaert, J.-P., Peng, T., Xiao, W., Pan, B., and Fang, X., 2022,

Dominant precessional forcing of the East Asian summer monsoon since 260 ka: *Geology*. Sun, Y., Yin, Q., Crucifix, M., Clemens, S. C., Araya-Melo, P., Liu, W., Qiang, X., Liu, Q., Zhao, H., Liang, L., Chen, H., Li, Y., Zhang, L., Dong, G., Li, M., Zhou, W., Berger, A., and An, Z., 2019, Diverse manifestations of the mid-Pleistocene climate transition: *Nature Communications*, v. 10, no. 1.

2. In fact, the precession cycle of the FeD/FeT is only obvious in a few time interval periods, such as 100-150 ka and 180-230 ka. Between 50-100 ka, 150-180 ka and 240-300 ka, the precession period is insignificant, instead the suborbital scale period is more obvious (See Figure 4).

3. For Fig. S8: It's strange that the wavelet and spectral analysis of proxies are not consistent. For example, the spectrum of Hm shows a dominant 100 ka cycle (d), but it's not shown in wavelet results (c), the spectrum result of δD displays dominant 100, 41 and 23 cycles (b), but the 100 ka cycle is not obvious in wavelet result compared to a significant 41-kyr cycle (a).

4. Again for the chronology. It's ok that the newly added OSL dating provides nice control for the upper part of the section. However, due to the lack of age controls in the lower part, the current correlation is not robust (the new correlation denoted by two black lines is also ok from my side). As to a paper that discusses the orbital cycles, the accuracy of the age frame is crucial. So, attempts should also be made to address this.

RESPONSE TO REVIEWERS

Reviewer #1 (Remarks to the Author):

The authors have satisfactorily addressed my concerns and I do not have further comments

We are grateful to the reviewer for acknowledging and recognizing our work, as well as for the valuable suggestions provided earlier.

Reviewer #2 (Remarks to the Author):

As I suggested in the last version, it would be important in solving the “Chinese 100-kyr Problem”, if the authors indeed present high-resolution independent temperature and precipitation records with accurate chronology. However, the current version cannot demonstrate such progress. The authors have tried to solve the concerns I proposed, for example, they added some new OSL dating points and re-correlated the original dating results with the Luochuan magnetic susceptibility (MS) records. However, the key of this study is how to demonstrate that the MS and DCB-extractable iron to total iron are independent temperature and precipitation proxies, respectively.

The main criticism is that the authors used a variety of proxies related to iron minerals to unmix the temperature and precipitation signals in an aeolian sequence. However, the proxy indication of iron minerals in the CLP and Madang may exhibit different controls, which may induce an unclear proxy indication. For example, magnetic susceptibility (MS) is used to reflect summer monsoon intensity in the CLP, and the MS curve can be well correlated throughout the CLP, thus providing a powerful tool for age control. The present study also finds that the Madang MS curve can match that in the CLP well, but the authors think that MS should reflect a dominant temperature response because in Madang there is abundant precipitation and pedogenesis is controlled by a major temperature effect. Such different proxy indications in different regions bring about difficulties to directly separating temperature and precipitation controls due to so close link between them in East Asia monsoon region.

We thank the reviewer for recognizing the significance of this work in potentially solving the "China 100-kyr problem". In this study, the MS curve was correlated with

that in the CLP to provide a precise age control. It is encouraging that the reviewer has agreed with this approach and accepted the timescale verified by independent absolute dating. The reviewer expressed concern that different proxy indications in different regions will bring about difficulties to our interpretation. In our opinion, these different proxy indications do not cause difficulties, but rather provide a rare opportunity to re-examine the indication of MS in the CLP. The MS has generally been used to reflect the intensity of the summer monsoon, but the temperature and precipitation controls have not been well separated. Our new observations and interpretations may provide important information to reinterpret the MS in the CLP. The Madang MS is interpreted as a temperature-dominated proxy, which is not in fundamental conflict with known evidence. The MS curve in the CLP can be well correlated with the brGDGTs-based MAT (Tang et al., 2017, see below for more details) and the benthic $\delta^{18}\text{O}$ curve. Obviously, the MS curve must carry a lot of information about the temperature change.

Moreover, a global temperature similar pattern does not mean direct temperature control. Given this, a direct temperature reconstruction based on biochemical proxy (e.g., GDGTs) may solve this problem.

We appreciate this constructive suggestion. We have analyzed both TOC and n-alkanes on pilot samples. The results show the TOC content of most samples is very low (0.24%) in comparison to the surface sample (1.37%), and the n-alkanes is below the limit of detection (Tab. 1). This fact indicates the decomposition of organic matter is severe in this low-latitudes. In this case the validity of GDGTs as a proxy of mean annual temperature (MAT) should be carefully checked by other studies in the future. However, the MAT has been estimated by the MAT_{mr} calibration based on 5- and 6-methylated brGDGTs for the loess-paleosol sequences of the Weinan section in the CLP (Tang et al., 2017). The results show that the MAT and MS curves in the Weinan section exhibit a largely synchronized variation over the past 350 kyr (Fig. S9). Hence, the following sentences were added into the text, which are “The mean annual temperature (MAT) was estimated using a biochemical proxy (brGDGTs) for the loess-paleosol sequences of the Weinan section in the CLP (Tang et al., 2017). The results show that the MAT and MS curves exhibit a largely synchronized variation over the past 350 kyr (Fig. S9), which implies that the MS is likely to be a temperature-dominated proxy.”.

Tab. 1 TOC and n-alkanes measured on pilot samples

Depth (m)	Mass (g)	TOC (%)	n-alkanes
0.05	19.74	1.37	D
0.60	19.90	0.29	ND
1.00	21.33	0.30	ND
1.85	20.05	0.49	ND
2.60	20.09	0.39	ND
3.15	19.74	0.17	ND
3.75	20.11	0.33	ND
4.45	19.84	0.12	ND
5.00	20.34	0.10	ND
6.15	19.93	0.36	ND
6.60	19.81	0.10	ND
7.55	19.83	0.13	ND
8.20	19.76	0.43	ND
8.60	20.08	0.11	ND
9.45	19.95	0.40	ND
10.20	20.26	0.13	ND
10.85	20.26	0.26	ND
11.85	19.72	0.08	ND
12.75	19.56	0.11	ND
13.40	20.17	0.31	ND
14.10	19.89	0.13	ND
14.50	19.89	0.12	ND
15.40	19.61	0.29	ND
15.90	20.16	0.34	ND

Note: D and ND denote Detectable and Non-detectable, respectively.

Fig. S9 Comparison of the MAT estimated by the MAT_{mr} calibration based on 5- and 6-methylated brGDGTs with the magnetic susceptibility (MS) for the loess-paleosol sequences of Weinan section in the CLP (Tang et al., 2017).

Reference

Tang, C. et al. Tropical and high latitude forcing of enhanced megadroughts in Northern China during the last four terminations. *Earth Planet. Sci. Lett.* **479**, 98–107 (2017).

In addition, I am strange that frequency-dependent magnetic susceptibility X_{fd} was not used in this study, which exhibits a clearer pedogenesis signal than MS.

In fact, we have measured the frequency dependent magnetic susceptibility (FDS). The result shows that both MS and FDS exhibit a strong correlation down the profile (correlation coefficient $r = 0.94$). We prefer to use MS because the repeat measurement shows some degree of drift in the FDS signals. We agree that FDS should be presented as supporting evidence. Therefore, we have included the FDS data in SI (Tab. S6, Fig. S6 and Fig. S7) and mentioned it in the text.

Fig. S6 was revised by adding the FDS time series

What's more, by comparing the temperature of Madang MS with that of the Antarctic ice core, the authors believe that MS is mainly controlled by temperature change. Obviously, such logic is problematic. A large number of studies have indicated that MS is controlled by the combined effects of both temperature and rainfall changes, so it is not surprising that there is a positive correlation with temperature (the R^2 value between them in this study is less than 0.4, this cannot be considered as a high correlation). The author does not rule out the influence of rainfall on MS here. If correlation analysis is conducted between MS and local rainfall, I think there may also be a positive correlation. In addition, the cross-correlation coefficient between MS and Hm is meaningless. In the Loess Plateau, MS also has a very good positive correlation with redness and hematite content, and even in the lake basin sediments of the NE Tibetan Plateau, such a high positive correlation exists, because the higher the redness, the higher the hematite or magnetic hematite content in the sedimentary records, and the corresponding MS value is higher.

Our discussion involves several interrelated steps of reasoning, as follows:

Step 1: The magnetic susceptibility (MS) may be influenced by the combined effects of both temperature and rainfall changes.

Step 2: It is observed that the MS and FeD/FeT characteristics of the Madang profile exhibit

different cycles related to different climate signals.

Step 3: Since FeD/FeT is proven to be precipitation-dominated, we deduce that the MS is not precipitation-dominated.

Step 4: The MS is positively correlated with temperature in the Antarctica and in the CLP and exhibits a 100-kyr cycle similar to the temperature record since the MPT.

Step 5: Additionally, previous studies have demonstrated that the hematite content of modern surface soil is primarily controlled by temperature.

Step 6: Given the correlation between the MS and hematite content in the Madang profile, it strongly suggests that the MS is predominantly influenced by temperature. Consequently, we consider the MS of the Madang profile to be temperature-dominated.

It is evident that our reasoning follows a rigorous line of argument.

In paleoclimate studies, correlation analysis is a crucial step in finding the potential link between different variables. Ideally, the Madang MS should be compared with known temperature and precipitation records. Unfortunately, there are no convincing precipitation records in this region. Madang and Antarctic are in different hemispheres and the proxy may be influenced by local factors. Given this, the observed correlation is significant enough. When FeD/FeT has been interpreted as a precipitation-dominated proxy, then MS, which varies in different cycles, must be interpreted as a proxy for another variable, i.e. temperature. Certainly, the influence of precipitation on MS cannot be ruled out, so we think the MS at Madang is only a temperature-dominated proxy.

How do the authors rule out the effect of temperature on the FeD/FeT indicator? In fact, the occurrence of unsynchronized changes in rain and heat at the orbital scale is very small in Madang region, which may not be consistent with the current meteorological record. Therefore, is it possible for the author to provide high-resolution palypollen records or other proxies that can directly reflect changes in temperature or rainfall to further support the views of this paper?

We believe that the FeD/FeT is also controlled by the combined effects of both temperature and precipitation changes. However, the relative contribution may vary from place to place. The contribution of temperature can be assessed from the record itself. If the FeD/FeT shows typical cycles of temperature records, such as 100-kyr or 40-kyr, it can be considered

temperature dominated. If the FeD/FeT does not show 100-kyr or 40-kyr cycles, it can be considered precipitation-dominated. We have examined the pollen of the Xiashu loess collected from different sites and found that the amount of pollen is extremely low compared to the standard required for meaningful statistics, e.g. 300 grains (Tab. R1). The poor preservation of pollen is attributed to the strong weathering at low latitudes. Therefore, no attempt was made to extract pollen from loess in Madang.

Tab. R1 Grain counts of pollen found in the Xiashu loess at different sites (Zhang et al., 2017)

Samples	ZJS-L1	LHT-L1	XC-L1	DG-S1	ZJS-S1	LHT-S1	ZJS-S5
Sum	1	4	17	3	2	4	4

Reference

Zhang W.C., et al. Pollen preservation and its potential influence on paleoenvironmental reconstruction in Chinese loess deposits. *Rev. Palaeobot. Palyno.* 240, 1–10 (2017).

Other concerns

1. For the introduction part, the authors did not describe the latest research progress well, and did not systematically summarize the main problems at present, and only cited the MS data of the LoessPlateau to elaborate, which is obviously one-sided.

For example, Guo et al.(2022) have suggested that the MS records in the CLP has a significant precession cycle, at least in the interglacial periods. The followings are some related paper.

Bao, R., Sheng, X., Meng, X., Li, T., Li, C., Shen, H., Da, J., Ji, J., and Chen, J., 2022, 100 k.y. pacing of the East Asian summer monsoon over the past five glacial cycles inferred from land snails: *Geology*, v. 51, no. 2, p. 179-183.

Guo, B., Nie, J., Stevens, T., Buylaert, J.-P., Peng, T., Xiao, W., Pan, B., and Fang, X., 2022, Dominant precessional forcing of the East Asian summer monsoon since 260 ka: *Geology*.

Sun, Y., Yin, Q., Crucifix, M., Clemens, S. C., Araya-Melo, P., Liu, W., Qiang, X., Liu, Q., Zhao, H., Liang, L., Chen, H., Li, Y., Zhang, L., Dong, G., Li, M., Zhou, W., Berger, A., and An, Z., 2019, Diverse manifestations of the mid- Pleistocene climate transition: *Nature Communications*, v. 10, no. 1.

We thank the reviewer for suggesting these references. The studies mentioned above and related to the loess brGDGTs (Tang et al., 2017) have been cited or discussed in the text. For example, the second paragraph of the was modified as follows:

Studies of the Chinese mid-latitude loess have produced many geological records of the EASM, of which those from the Chinese Loess Plateau are the most classic (An et al., 1990, 1991; Ding et al., 1992; Guo et al., 1994). The ~~proxy-indicator-of~~ magnetic susceptibility (MS) reflects the dominant 100-kyr cycle of the EASM since the Middle Pleistocene (An et al., 1990, Liu and Ding, 1998; Lu et al., 1999; Sun et al., 2006; Beck et al., 2018; Lu et al., 2022). Both $\delta^{13}\text{C}$ of inorganic carbonate (Sun et al., 2019) and $\delta^{13}\text{C}$ of land snail shells (Bao et al., 2022) exhibit the same cycle. Because the strongest EASM recorded on the Chinese Loess Plateau appeared during an interglacial period, which was characterized by a low global ice volume, the change in the EASM is thought to be driven by global ice volume variability (Liu and Ding, 1998; Sun et al., 2019; Bao et al., 2022). Unlike the proxies mentioned above, a new proxy composited by sand content and MS reveals a dominant 23-kyr cycle caused by Northern Hemisphere summer insolation forcing (Guo et al., 2022).

2. In fact, the precession cycle of the FeD/FeT is only obvious in a few time interval periods, such as 100-150 ka and 180-230 ka. Between 50-100 ka, 150-180 ka and 240-300 ka, the precession period is insignificant, instead the suborbital scale period is more obvious (See Figure 4).

It is true that FeD/FeT has not followed the precessional change in some intervals. Several reasons for this discrepancy have been suggested in the discussion. Constrained by the relatively low resolution of the present timescale, it is inappropriate to address the suborbital cycles, especially when they do not reach the 95% confidence level.

3. For Fig. S8: It's strange that the wavelet and spectral analysis of proxies are not consistent. Foreexample, the spectrum of Hm shows a dominant 100 ka cycle (d), but it's not shown in wavelet results (c), the spectrum result of δD displays dominant 100, 41 and 23 cycles (b), but the 100 ka cycle is not obvious in wavelet result compared to a significant 41-kyr cycle (a).

Since the Madang record is not long enough, the resulting 100-kyr cycle is only visible

in the lowermost part (indicated by the red arrows). The thick black contour indicates the 5% significance level against red noise, and the cone of influence (COI), where edge effects can distort the image, is shown as a lighter shade.

The 100-kyr cycle (indicated by red arrows) is not obvious in the wavelet transform plot due to the edge effects

4. Again for the chronology, I am happy to see that the newly added OSL dating provides nice control for the upper part of the section. However, due to the lack of age controls in the lower part, the current correlation is not robust (the new correlation denoted by two black lines is also ok from my side). As to a paper that discusses the orbital cycles, the accuracy of the age frame is crucial. So, attempts should also be made to address this.

We are encouraged by the reviewers' recognition of our additional OSL dating in the upper profile. However, the suggested correlation, indicated by two black lines, will result in a mismatch, i.e., the circled MS peak of Luochuan is correlated with the circled MS peak of Madang. It is obvious that the circled MS peak of Luochuan appears in a glacial period (to the left of the dashed red lines), and the circled MS peak of Madang appears in an interglacial period (to the left of the dashed red lines). We also performed

spectral analysis on the new time series obtained using the suggested correlation. The results show that both the 40-kyr cycle of MS and the 18-kyr cycle of FeD/FeT are no longer detectable. Therefore, we prefer our correlation scheme. It is clear that climatostratigraphy can provide reliable timescale, such as the timescale established for loess in the CLP.

The suggested correlation (black lines) will introduce a mismatch by linking a glacial MS peak (Luochuan) to an interglacial MS peak (Madang)

The results show both the 40-kyr cycle of MS and 18-kyr cycle of FeD/FeT are no

longer detectable

For age control in the lower part of the profile, we attempted to validate the timescale by measuring OSL ages using feldspar grains. However, this approach failed because the feldspar ages of two pilot samples deviated significantly from the expected ages (Table R2). This deviation will further increase the age error, which generally increases with age over the last three glacial-interglacial cycles in the CLP (Zhang et al., 2022). It is clear that the resulting error does not allow the establishment of a timescale that meets the requirements for spectral analysis. For this reason, we have not performed feldspar OSL dating for samples older than the last interglacial. The fine-grained feldspar has been altered by weathering at this low latitude, and this is thought to be the cause of the deviation.

Tab. R2 OSL ages for fine-grained feldspar of the Madang profile

Sample	U (ppm)	Th (ppm)	K (%)	W.C. (%)	De (Gy)	Aliqo uts Num.	Dose rate (Gy/ka)	Age (ka)	Expect ed age (ka)
MD-P- 1.5m	3.21±0. 12	13.90±0 .38	1.71±0. 05	17.4	425± 33	7	4.11±0. 39	103.4±1 2.6	75.9
MD-P- 4.5m	2.44±0. 10	15.40±0 .42	1.96±0. 06	17.6	574± 67	3	4.19±0. 39	137.0±2 0.5	102.3

Once again, we would like to express our sincere gratitude to both reviewers for their highly professional comments and valuable suggestions during the review process. We have worked diligently to improve the quality of the manuscript and hope that the revisions and responses we have made are recognized and accepted by the reviewers.

Reviewer #3 (Remarks to the Author):

Please see attached.

Reviewer #3 Attachment on the following page

In their review of the third version of this manuscript, Reviewer #3 added some comments to the manuscript file. These comments were forwarded to the authors, who replied as included in this Peer Review File.

Dear editor and author team,

Thanks for the opportunity to read this interesting manuscript. I was called on to evaluate this manuscript after two rounds of reviews, which seem to yield conflicting opinions (I didn't see the reviewer comments from the first round) regarding this manuscript. After reading the manuscript several times along with comments made by the reviewers during the second round, I feel that the reviewers have made great suggestions and this manuscript has been improved greatly because the authors have done substantial work to incorporate reviewers' comments. To me, this manuscript has merits and the FeD/FeT ratio record from southern China loess deposits is very valuable to promote understanding East Asian summer monsoon (EASM) cyclicities and dynamics, which has attracted much attention in geosciences and are some of the most perplexing problems in monsoon studies. Therefore, the question this research aims to resolve has broad implications which are of interests of general audience. This research is novel too because past efforts focusing on understanding EASM orbital cyclicities mainly focused on loess deposits on the Chinese Loess Plateau (CLP) and the speleothem records in East Asia. Recently, studies have been performed using Fe-related parameters to extract dry-wet variations from southern China eolian deposits, but the results were different from this one, presumably because the site chosen here is superior (high elevation, well-drained loess). Furthermore, Liu et al. (2021) tried to evaluate EASM orbital cyclicities from the Tengger desert site, which provides new insight into this classic question, but Tengger desert is located in monsoonal marginal region and whether results there can be applied to infer monsoon variations on the CLP is unknown. Within this context, the results presented here is timely and it offers a rare opportunity to reevaluate EASM cyclicities and dynamics from a broad region based on eolian sediments: Tengger, CLP, and southern China. Therefore, the research design is unusual and worth applauding and the results are very welcome and timely.

The strength of this work: the authors use modern soil calibration from southern China to demonstrate FeD/FeT is sensitive to precipitation variations but not to temperature. Therefore, the FeD/FeT records of the chosen site should be able to resolve precipitation variations (the site is unique because it is located in a high topography area with well-drained soils, so it is devoid of potential groundwater effect, similar to the CLP sites). Within this context, the team demonstrates that the EASM has dominant 20-kyr band cyclicities from EASM core area for the first time using archives other than speleothem. This is an intriguing finding in my opinion, no matter whether their interpretation regarding magnetic susceptibility (MS) is correct or not, which is one debate focus in previous reviews and responses. Bundling modern calibration data with paleo record interpretation, contrasting magnetic proxy records with FeD/FeT proxy records, comparing CLP with southern China data, are some of novel nature that this research has, which have stimulated me to think deeper about how to get a coherent interpretation of each proxy from broad research areas. So I thank the authors for this work and what they have achieved so far to get this published in a high profile journal.

The portions that can be improved:

Interpretation for MS variations: I feel that the interpretation for MS variations in their site needs more careful analysis. And alternative interpretations should be explored. I stated so based on a few reasons. First, some workers researching loess magnetic enhancement mechanism did

propose MS enhancement may be determined by air or soil temperature (for example Torrent et al., GRL), but this is speculative and it was based on high temperature experiments in laboratories that may not be directly applicable to room temperature conditions as in natural environments. On the other hand, most other studies exploring loess MS enhancement mechanisms link MS enhancement to precipitation variations at least for the CLP area where precipitation is a limiting factor. Therefore, linking MS variations to be primarily controlled by temperature still lacks solid theoretical basis at this stage. Second, the dominant 100-kyr cycles on the CLP, and maybe the study site, could be caused by signal smoothing caused by postdepositional processes (see comments from the annotated file). This possibility should be discussed in the revised version. Third, the temperature interpretation seems to be in conflict with the observation that the examined southern China site loess MS have lower values than those from the CLP. Obviously, it is reasonable to assume that southern China should have been warmer than the CLP in the late Quaternary. On the other hand, my suggested alternative interpretation regarding the MS variations (signal smoothing) would be able to circumvent this drawback.

Incorporating more most updated research from a more broad area focusing on EASM: Indeed, instead of ignoring/criticizing loess records on the CLP, if the authors include newest progresses on the CLP (from different parts of the plateau; to highlight some: Stevens et al., Chen et al., Guo et al, Sun et al.) and the Tengger desert records and compared them with the FeD/FeT record reported here from southern China, they may be able to make a more persuasive case in demonstrating the major role of precessional cycles on EASM precipitation. This effort would make this research stand on the cutting edge of paleo- EASM research. I tried to compare the FeD/FeT data generated in this study with the monsoon stack data generated by Guo et al., and the comparison shows lots of similarities (see the plot following this letter).

Age model: Using orbital tuned age model is a widely used strategy in paleoceanography research, and I suppose that this strategy can be applied to loess research here, which have been done many times before for the CLP sites. Particularly, when tuning was done using one proxy record but other proxy records show different cycles as the proxy record used for tuning (sort of a similar case here). See Liu Zhonghui et al. Nature 2004 for an example, where age model was established by tuning benthic oxygen isotope record, but the authors focused on analyzing orbital cycles from alkenone SST and productivity data and variations of SST and productivity show different responses as the benthic oxygen isotope ratios. This may be one way to alleviate the potential critique of circular reasoning in terms of age model.

I do agree with the reviewer that it is best to combine orbital tuning with absolute dating, which seems to be the strategy used by this manuscript after incorporating suggestions by reviewers. However, I think uncertainties of the preferred age model are better to be highlighted and different possibilities are better to be included in the manuscript. Currently, an alternative age model was suggested by the reviewer and the authors consider this possibility and exclude it only in the response letter. I would suggest that they include this alternative age model as supplementary materials and discuss the potential implications if this alternative age model is correct. This way, this manuscript would be in a safer ground.

In summary, I agree with many of the critiques of the reviewers made, which improved the robustness of this manuscript in many ways. However, I feel that with more careful treatments, all of these suggestions can be incorporated and addressed, especially when they incorporate

more recent monsoon records of the similar mentality (from new sites, using new proxies, using more in-depth analysis of existing proxies). Together, I believe that they would be able to make a compelling case which would have potentials of revolutionize past thinking regarding EASM variations and dynamics.

Junsheng Nie

RESPONSE TO REVIEWERS (highlighted in blue)

Dear editor and author team,

Thanks for the opportunity to read this interesting manuscript. I was called on to evaluate this manuscript after two rounds of reviews, which seem to yield conflicting opinions (I didn't see the reviewer comments from the first round) regarding this manuscript. After reading the manuscript several times along with comments made by the reviewers during the second round, I feel that the reviewers have made great suggestions and this manuscript has been improved greatly because the authors have done substantial work to incorporate reviewers' comments. To me, this manuscript has merits and the FeD/FeT ratio record from southern China loess deposits is very valuable to promote understanding East Asian summer monsoon (EASM) cyclicities and dynamics, which has attracted much attention in geosciences and are some of the most perplexing problems in monsoon studies. Therefore, the question this research aims to resolve has broad implications which are of interests of general audience. This research is novel too because past efforts focusing on understanding EASM orbital cyclicities mainly focused on loess deposits on the Chinese Loess Plateau (CLP) and the speleothem records in East Asia. Recently, studies have been performed using Fe related parameters to extract dry wet variations from southern China eolian deposits, but the results were different from this one, presumably because the site chosen here is superior (high elevation, well drained loess). Furthermore, Liu et al. (2021) tried to evaluate EASM orbital cyclicities from the Tengger desert site, which provides new insight into this classic question, but Tengger desert is located in monsoonal marginal region and whether results there can be applied to infer monsoon variations on the CLP is unknown. Within this context, the results presented here is timely and it offers a rare opportunity to reevaluate EASM cyclicities and dynamics from a broad region based on eolian sediments: Tengger, CLP, and southern China. Therefore, the research design is unusual and worth applauding and the results are very welcome and timely.

We sincerely thank the reviewer for the high praise and enthusiastic encouragement of our work. To our understanding, the Madang profile in this study is indeed a very rare and valuable material because it is a high-resolution loess record from the core zone of the East Asian Summer Monsoon. As the reviewer pointed out, it provides a rare opportunity to re-evaluate EASM cyclicities and dynamics from a broad region based on eolian deposits.

The strength of this work: the authors use modern soil calibration from southern China to demonstrate FeD/FeT is sensitive to precipitation variations but not to temperature. Therefore, the FeD/FeT records of the chosen site should be able to resolve precipitation variations (the site is unique because it is located in a high topography area with well drained soils, so it is devoid of potential groundwater effect, similar to the CLP sites). Within this context, the team demonstrates that the EASM has dominant 20 kyr band cyclicities from EASM core area for the first time using archives other than speleothem. This is an intriguing finding in my opinion, no matter whether their interpretation regarding magnetic susceptibility (MS) is correct or not, which is one debate focus in previous reviews and responses. Bundling modern calibration data with paleo record interpretation, contrasting magnetic proxy records with FeD/FeT proxy records, comparing CLP with southern China data, are some of novel nature that this research has, which have stimulated

me to think deeper about how to get a coherent interpretation of each proxy from broad research areas. So I thank the authors for this work and what they have achieved so far to get this published in a high profile journal.

Thanks for recognition.

The portions that can be improved:

Interpretation for MS variations: I feel that the interpretation for MS variations in their site needs more careful analysis. And alternative interpretations should be explored. I stated so based on a few reasons. First, some workers researching loess magnetic enhancement mechanism did propose MS enhancement may be determined by air or soil temperature (for example Torrent et al., GRL), but this is speculative and it was based on high temperature experiments in laboratories that may not be directly applicable to room temperature conditions as in natural environments. On the other hand, most other studies exploring loess MS enhancement mechanisms link MS enhancement to precipitation variations at least for the CLP area where precipitation is a limiting factor. Therefore, linking MS variations to be primarily controlled by temperature still lacks solid theoretical basis at this stage. Second, the dominant 100-kyr cycles on the CLP, and maybe the study site, could be caused by signal smoothing caused by postdepositional processes (see comments from the annotated file). This possibility should be discussed in the revised version. Third, the temperature interpretation seems to be in conflict with the observation that the examined southern China site loess MS have lower values than those from the CLP. Obviously, it is reasonable to assume that southern China should have been warmer than the CLP in the late Quaternary. On the other hand, my suggested alternative interpretation regarding the MS variations (signal smoothing) would be able to circumvent this drawback.

To address these concerns, we have made some necessary changes to the manuscript as follows:

(1) The ~20-kyr cycle of precipitation, as indicated by the low-latitude loess FeD/FeT, has been emphasized more prominently, addressing a key aspect of the "*Chinese 100-kyr cycle problem*" controversy. The manuscript's title has also been revised to "Loess Deposits Reveal the ~20-kyr Cycle of Precipitation Variations in the Low Latitudes of East Asia Since ~350 ka."

(2) The ~100-kyr temperature cycle, as indicated by Madang's magnetic susceptibility, has been given a secondary emphasis, and the term "may be" has been incorporated to convey our speculation rather than a definitive conclusion. For instance, in the Abstract, we revised it to: "*Our analyses reveal that variations in the ratio of DCB extractable iron to total iron are dominated by a ~20 kyr cycle, reflecting changes in precipitation. In contrast, magnetic susceptibility varies with a ~100-kyr cycle and may be mainly influenced by temperature variations.*"

(3) Two recent monsoon records from the western CLP (Guo et al., 2022) and the Tengger Desert (Liu et al., 2021) have been integrated into the discussion, and a comparative diagram has been included in the Supplementary Information (Fig. S13).

(4) We have also included the findings of a recent simulation study of the EASM in the revised manuscript. The simulation results lend support to our research conclusions, indicating a dominant ~20-kyr cycle of precipitation and a distinct ~100-kyr cycle of temperature in the Yangtze River Valley (Dai et al., 2022).

(5) We have explored further into the climatic significance and potential mechanisms of the FeD/FeT ratio and MS. The enhancement of MS is attributed to the oxidation of magnetite, primarily influenced by ambient temperature. In contrast, FeD/FeT is sensitive to leaching, showing a high dependence on rainfall in southern China.

We would like to emphasize that this manuscript exclusively focuses on proxy indications in low latitudes. The evidence we have obtained does not allow us to comment on previous studies conducted in middle latitudes. We can only provide a self-consistent explanation for the observed phenomena in Madang. Each proxy indicator has its distinct range of application. In simpler terms, a proxy indicator in a different region may be sensitive to different climatic factors. We have not dismissed the indicative significance of MS in middle latitudes, but have raised awareness of the need to reassess its climatic significance. In fact, this reassessment has been continuously attempted by many researchers.

The smoothing caused by postdepositional processes may offer a possible explanation for the absence of a dominant ~20-kyr cycle in the MS record from the central CLP. Postdepositional processes, such as MS-related pedogenesis or FeD/FeT-related leaching, are likely involved in the downward smoothing. If the enhancement of MS is a result of magnetite oxidation and FeD is generated by the dissolution of iron-bearing silicate minerals, it appears that leaching could lead to more pronounced smoothing. In fact, the FeD/FeT records a shorter cycle than the MS. Additionally, we have examined the grain size and observed that it is synchronous with the MS down the profile in the CLP.

The temperature interpretation for Madang MS is not in conflict with the observation, as lower MS in southern China is likely a result of the suppression of excessive precipitation, as revealed by many studies on modern surface soil.

Incorporating more most updated research from a more broad area focusing on EASM: Indeed, instead of ignoring/criticizing loess records on the CLP, if the authors include newest progresses on the CLP (from different parts of the plateau; to highlight some: Stevens et al., Chen et al., Guo et al., Sun et al.) and the Tengger desert records and compared them with the FeD/FeT record reported here from southern China, they may be able to make a more persuasive case in demonstrating the major role of precessional cycles on EASM precipitation. This effort would make this research stand on the cutting edge of paleo-- EASM research. I tried to compare the FeD/FeT data generated in

this study with the monsoon stack data generated by Guo et al., and the comparison shows lots of similarities (see the plot following this letter).

We appreciate the reviewer for this constructive suggestion. New records from the monsoon marginal zone have been incorporated into discussion. For instance, we incorporated the sentences into the manuscript as following: “Two records from the monsoon marginal zone show a cyclic variation similar to the East Asian speleothem record over the past ~300 kyr (Liu et al., 2021; Guo et al., 2022). A record of EASM precipitation from the western CLP shows a dominant cycle of 23 kyr over the past 260 kyr (Guo et al., 2022), while another record of wet–dry variations from the Tengger Desert has a clear 20-kyr cycle over the past 400 kyr (Liu et al., 2021). These are similar to the dominant 18-kyr cycle identified in this study, suggesting that precipitation (wetness) has varied synchronously in the monsoon zone, with high precipitation (wetness) generally correlated with high solar insolation (Fig. S13).”

Age model: Using orbital tuned age model is a widely used strategy in paleoceanography research, and I suppose that this strategy can be applied to loess research here, which have been done many times before for the CLP sites. Particularly, when tuning was done using one proxy record but other proxy records show different cycles as the proxy record used for tuning (sort of a similar case here). See Liu Zhonghui et al. Nature 2004 for an example, where age model was established by tuning benthic oxygen isotope record, but the authors focused on analyzing orbital cycles from alkenone SST and productivity data and variations of SST and productivity show different responses as the benthic oxygen isotope ratios. This may be one way to alleviate the potential critique of circular reasoning in terms of age model.

Appreciation is extended for the reviewer's recognition and suggestion. In the revised manuscript, we have incorporated the information and reference mentioned by the reviewer into the discussion section concerning the establishment of the age framework. “Using an orbital-tuned age model is a widely adopted strategy in paleoceanography research. This method is equally applicable to the loess research, as evidenced by its successful application in various studies involving CLP sites. Specifically, this approach proves effective when tuning is conducted with one proxy record, while other proxy records reveal cycles distinct from the one used for tuning (Liu and Herbert, 2004).”

I do agree with the reviewer that it is best to combine orbital tuning with absolute dating, which seems to be the strategy used by this manuscript after incorporating suggestions by reviewers. However, I think uncertainties of the preferred age model are better to be highlighted and different possibilities are better to be included in the manuscript. Currently, an alternative age model was suggested by the reviewer and the authors consider this possibility and exclude it only in the response letter. I would suggest that they include this alternative age model as supplementary materials and discuss the potential implications if this alternative age model is correct. This way, this manuscript would be in a safer ground.

Yes, we adopted a combined approach of magnetic susceptibility stratigraphic correlation (compared with the Luochuan profile stratigraphic age established through orbital tuning) and absolute dating for the establishment of the age model. Because the MS variation curves of the two profiles exhibit striking similarity, we consider the chosen nine depth-age tie points to be reliable. The previous reviewer proposed another age model. However, this would conflict with the stratigraphic correlation, i.e. it would correlate the Luochuan loess unit with the Madang palaeosol unit. Due to this discrepancy, the suggested age model was not considered as a viable alternative.

In summary, I agree with many of the critiques of the reviewers made, which improved the robustness of this manuscript in many ways. However, I feel that with more careful treatments, all of these suggestions can be incorporated and addressed, especially when they incorporate more recent monsoon records of the similar mentality (from new sites, using new proxies, using more in-depth analysis of existing proxies). Together, I believe that they would be able to make a compelling case which would have potentials of revolutionize past thinking regarding EASM variations and dynamics.

We thank the reviewer for his encouragement. Additional recent monsoon records were incorporated into the manuscript, and the proxy indications were further discussed. In addition to the two recent monsoon records from the western CLP (Guo et al., 2022) and the Tengger Desert (Liu et al., 2021), we have also incorporated the latest simulation results of the EASM into the revised manuscript (Dai et al., 2022): *“Temperature and precipitation in East Asia during the past 425 ka have recently been simulated. The simulated MAT in East Asia is mainly dominated by the 100-kyr cycle. The simulated MAP shows a distinct 40-kyr cycle in North China, 20- and 40-kyr cycles in the Yangtze River Valley, and a 20-kyr cycle in South China. This simulation suggests that the importance of the 20-kyr cycle increases gradually from north to south in the East Asian monsoon region (Dai et al., 2022). This result further supports our interpretation that the dominant 100-kyr and 20-kyr cycles in the Yangtze River Valley reflect temperature and precipitation, respectively. Unlike the simulated MAP in the Yangtze River Valley, our record does not show the 40-kyr cycle.”*

Commented [A1]: On the central CLP. For western CLP, both precessional and semiprecessional cycles have been detected.

These studies were added.

Commented [A2]: Note this statement is not so precise: the cited reference present a magnetic susceptibility record generated by Chen Fahu’s group from the western CLP which also show dominant 20 kyr cycles! This magnetic susceptibility record showing dominant 20kyr cycles paves the way for the stacking in order to generate longer records.

This sentence was changed into *“a spliced EASM record reveals a dominant 23-kyr cycle caused by Northern Hemisphere summer insolation forcing (Guo et al., 2022).”*

Commented [A3]: Terminology usage needs to be more carefully examined. From what I learnt, Dr. Cheng Hai often attributed delta18O of speleothems to general circulation, which may or may not be equal to precipitation.

We really know the latest expression of Dr. Cheng Hai. Here, we followed the original expression of the articles. To avoid causing any confusion, we have removed the "precipitation".

Commented [A4]: This is correct expression, I think.

Thanks for the recognition.

Commented [A5]: But you said above this indicate monsoon intensity. Is intensity equal to precipitation? I am a bit confused with the terminology used here.

To prevent any potential misunderstanding and confusion, we have also omitted "precipitation" from this sentence, despite the original references conflating East Asian summer monsoon and precipitation. For example, in this paper “Yuan et al., Timing, Duration, and Transitions of the Last Interglacial Asian Monsoon. *Science* 304, 575-578(2004)”, the expressions are not consistent (please see copies listed below).

Thorium-230 ages and oxygen isotope ratios of stalagmites from Dongge Cave, China, characterize the Asian Monsoon and low-latitude precipitation over the past 160,000 years. Numerous abrupt changes in $^{18}\text{O}/^{16}\text{O}$ values result from

(25). Indeed, the low-frequency component of the whole Hulu/Dongge record (Fig. 3) correlates with insolation, indicating that insolation is important in controlling monsoon intensity as predicted (25). At higher frequen-

Thus, the Hulu/Dongge record indicates major and abrupt changes in tropical and subtropical precipitation, which correlate with

Commented [A6]: Intensity or precipitation or both or intensity is equal to precipitation?

Here, EASM can refer to both intensity and precipitation. Strictly, EASM intensity is not exactly equal to EASM precipitation. However, there is a significant correlation between EASM intensity and EASM precipitation, especially when discussing the variation cycles of EASM.

Commented [A7]: See discussion in Guo et al. 2022 *Geology* for different opinions regarding simulations.

Here, we have simply listed speculations on several potential factors related to the occurrence of the "Chinese 100-kyr problem." Regarding simulations, Guo et al. 2022 states, “Some model simulations suggest that the EASM did not exhibit strong precessional cycles over the middle to late Quaternary (Liu et al., 2014; Cheng et al., 2021), but these results are not consistent with those of other model simulations (Lyu

et al., 2021). Furthermore, no previous model simulations have been performed specifically focused on the CLP area, and these simulations have room for improvement. Our results call for high-resolution model simulations specifically focusing on the CLP area.” We generally concur with this explanation.

Commented [A8]: How can Sr/Ca ratio reveal insights into processes controlling MS variations? It is not so straightforward to understand. Please elaborate.

The sentence directly cited the statements in the original (please see copies listed below).

By extending the existing database, we show that microcodium Sr/Ca ratio is more reliable as a summer precipitation proxy on the Chinese Loess Plateau because of its weak dependence on temperature, calcification rate, and composition of the original soil solution compared with Mg/Ca ratio. The $\delta^{18}\text{O}$ values of Holocene microcodium indicate

kyrs pacing after the mid-Pleistocene transition (Fig. 7b). Whether or not the 100-kyr glacial–interglacial cycles are driven by obliquity or eccentricity (Huybers and Tziperman, 2008), the patterns are very different from the Sr/Ca summer precipitation records at the exact same sites. This argues that most of the current proxies that trace the loess-and-paleosol alternations is influenced by other processes than the EASM. The

Li, T. et al. Continued obliquity pacing of East Asian summer precipitation after the mid-Pleistocene transition. *Earth Planet. Sci. Lett.* 457, 181–190 (2017).

Commented [A9]: I think there is another potential factor that could reconcile the inconsistency, as highlighted by Guo et al. 2022: MS can record monsoon precipitation during the interglacials but not during glacials because of diminished precipitation over glacials on the CLP: this may have caused precipitation below the threshold value for magnetic enhancement to occur. This factor is suggested to be added because it is different from any of the factors listed here.

We revised as suggested. The opinion (Guo et al., 2022) was added here as another possible explanation. Additionally, the smoothing effect of postdepositional processes on the MS signal has also been listed here as a potential factor. “(5) postdepositional processes smoothed the MS signal, increasing the 100- kyr cycles and decreasing the 20- and 40- kyr cycles (Nie et al., 2008); and (6) reduced precipitation during glacials on the CLP might have fallen below the threshold for magnetic enhancement, preventing the recording of monsoon precipitation by MS during glacials, although it remains observable during interglacials (Guo et al., 2022)”.

Commented [A10]: Necessary is a strong word. I would use word like “useful”. This is actually a smart idea: looking at monsoon variations from outside the classic CLP research sites. Actually, Liu Chengying et al. used a similar strategy to generate monsoon records from the area north of the CLP. Maybe it is a good idea to introduce the work by Liu Chengying et al. after this sentence, as her work also detected strong precessional cycles (their Fig. 3), which are consistent with the results of this manuscript over the similar interval. Comparing results with Liu et al. would lend more confidence to the work reported here. Otherwise, the introduction did not incorporate some newest findings regarding monsoon variations, as is pointed out by the other reviewers.

This is a valuable suggestion, and we revised as suggested. “To determine the cause of the “Chinese 100-kyr problem”, it is useful to obtain geological records far away from the CLP. Two records from the monsoon marginal zone show a wet–dry variation similar to that of the East Asian speleothem over the past ~300 kyr (Liu et al., 2021; Guo et al., 2022). However, records from the monsoon core zone are still lacking. Therefore, the loess in the lower reaches of the Yangtze River was chosen as the subject in this study (Fig. 1).”

Commented [A11]: Why does this matter? I did not follow.

This sentence was deleted.

Commented [A12]: How thick is the Luochuan loess used to be compared with the loess record here, for the same time interval?

Luochuan profile down to 16.29 m depth (369 ka) was used.

Commented [A13]: This is more commonly shortened as CBD in my impression.

Both abbreviations are used in literatures. Pedological articles may prefer DCB.

Commented [A14]: Some description about the results would be best to be included in the main text to guide understanding.

“The obtained OSL ages are consistent with stratigraphic sequence, ranging from 30.1 ± 2.2 ka (at 40 cm depth) to 85.6 ± 6.0 ka (at 350 cm depth). The time series of MS defined by OSL ages can be well correlated with that of Luochuan.” The above sentences were added in the text.

Commented [A15]: This does not belong to Results and can be deleted here.

This sentence was deleted.

Commented [A16]: It is interesting to notice that Luochuan MS is higher than Madang MS in general. Any potential reason for this pattern?

It is true that the MS is relatively low in South China. A study on modern surface soils has found that the MS in South China decreases with increased temperature and precipitation (Han et al., 1996). In the revised manuscript, we have added a paragraph addressing this phenomenon and attempted to provide explanations for possible causes: “Notably, there is evidence that MS in Madang may also be influenced by precipitation. For example, the MS in Madang is generally lower than that in Luochuan. Excessive precipitation may suppress MS because the MS of modern soils in southern China decreases with increasing precipitation (Han et al., 1996).”

Redacted

Han J., Lyu H., Wu N., et al. The magnetic susceptibility of modern soils in China and its use for paleoclimate reconstruction. *Stud. Geophys. Geod.* **40**, 262–275 (1996).

Commented [A17]: I wonder why there is this difference

The revised passage may provide an explanation for the observed difference. "A study of modern soils with different parent materials along a north–south climatic gradient in eastern China indicated positive relationships between FeD and mean annual precipitation (MAP), as well as mean annual temperature (MAT). However, an obvious increase in FeD is only observed when MAP exceeds ~800 mm, and correspondingly, MAT is higher than ~15 °C (Fig. S10; Peng et al., 2022). This may be the reason why the changes in FeD/FeT occurred on the CLP without an obvious precession cycle. The modern MAP and MAT on the CLP are low; for example, only 592 mm and 9.6 °C at Luochuan, respectively. Under high rainfall regimes, such as in southern China, the enrichment of FeD controlled by chemical weathering is favoured by an increase in rainfall when it does not exceed the specified rainfall threshold (Long et al., 2016). A study conducted in South China revealed a strong positive correlation between FeD/FeT and MAP when MAP ranges from 900 mm to 1720 mm. However, no correlation was found between FeD/FeT and MAT in this study (Fig. S11; Fig. S12; Long et al., 2016). It seems that FeD/FeT is a precipitation-dominated proxy indicator in low-altitude areas."

Commented [A18]: Or they have different sensitivities to postdepositional disturbance. MS is very sensitive to the ultrafine ferrimagnetic grains, by contrast FeD/FeT have two terms and are controlled by content of both free iron and total iron in soils. In southern China, free iron content variations may be much larger than those on the CLP due to wetter climate. Therefore MS and FeD/FeT have different cyclicities between the CLP and southern China loess. Indeed, the fact that MS has lower values in general for the examined southern China site than Luochuan is an indication that wet climate in southern China may be not favorable for ultrafine ferrimagnetic grains to form.

Yes, we agree with both points: (1) In southern China, variations in FeD may be significantly larger than those on the CLP due to the wetter climate. As a result, FeD/FeT displays different cyclicities between the CLP and southern China loess; (2)

the wetter climate in southern China may not be conducive to the formation of ultrafine ferrimagnetic grains.

The following sentences has been incorporated into the revised manuscript.

“The enhancement of MS and FeD/FeT may be related to different postdepositional processes, with MS being sensitive to the oxidation of ultrafine magnetite grains (Nie et al., 2010) and FeD/FeT being sensitive to leaching. Oxidation of magnetite in air is primarily influenced by the ambient temperature (Nasrazadani and Raman, 1993). The FeD/FeT ratio mainly depends on the accumulation rate of pedogenic iron oxides relative to primary iron-bearing silicate minerals (Long et al., 2016). Increased rainfall leads to more pronounced leaching, resulting in a greater dissolution of silicate minerals. This process releases more pedogenic iron oxides, facilitating their migration to and accumulation in the B horizon (or AB horizon) of the soil (Lei et al., 2001), ultimately leading to an increase in FeD/FeT. Therefore, MS and FeD/FeT are likely to be sensitive to temperature and precipitation, respectively.”

Commented [A19]: See also Nie et al., 2010 and Song Yang et al. 2014 for different opinions.

We have noted the new opinion on the enhancement mechanism of Yang et al. (2014) suggested that annual rainfall rather than temperature exerts the dominant effect on soil magnetic enhancement. However, the PCA analysis shows no obvious difference between MAT (0.751) and MAP (0.925). It has been suggested that magnetic enhancement is related to the ultrafine pedogenic maghemite grains derived from oxidation of ultrafine magnetite grains (Nie et al., 2010). It is clear that the higher temperature not the higher precipitation benefits the oxidation. Therefore, this study was cited in the revised manuscript. “The enhancement of MS and FeD/FeT may be related to different postdepositional processes, with MS being sensitive to the oxidation of ultrafine magnetite grains (Nie et al., 2010) and FeD/FeT being sensitive to leaching.”

Song Y., Hao Q., Ge J., et al. Quantitative relationships between magnetic enhancement of modern soils and climatic variables over the Chinese Loess Plateau, *Quatern. Int.* **334–335**, 119-131 (2014).

Nie, J., Song, Y., King, J. W., et al. Consistent grain size distribution of pedogenic maghemite of surface soils and Miocene loessic soils on the Chinese Loess Plateau. *J. Quaternary Sci.* **25**, 261–266 (2010).

Commented [A20]: The absolute values are lower for the southern site. This needs to be explored a bit.

Please refer to the response provided for Comment [A16].

Commented [A21]: Or both affected by postdepositional processes to a similar degree.

“or that these sites have undergone similar postdepositional processes” was added here.

Commented [A22 and A23]: One problem with the MS records on the CLP is that the western sites and central sites show different periods. Also, one recent loess MS record from northern CLP with absolute ages by Tom Stevens and coauthors reveal clear precessional cycles. So instead of attributing the MS record (used here from central CLP) to temperature control, maybe the author team can argue using one way that I have proposed before: MS records from the central CLP may not be able to record real cyclicities of forcing because bioturbation and other post-depositional processes have caused signal smoothing, which enlarged the 100kyr cycles and diminished the smaller 20 and 40 kyr band cycles (Nie et al., 2008 GSAB; Guo et al., 2022). After all, MS is sensitive to ferrimagnetic minerals' grain size and concentration, but the absolute amount of ferrimagnetic minerals in soils are limited and small changes can affect them in a great way. However, total Fe in soils have high concentrations and postdepositional processes would in theory leave a smaller degree of impact on them. To me, this is another way, if not more reasonable, to explain why FeD/FeT can record more apparent smaller orbital cycles than MS. This way, I feel that the interpretations may be able to satisfy the other reviewers and are more consistent with current understanding of climatic implications of loess MS as well.

It is not our original intention to be involved in interpreting the MS indication in CLP. Now that the reviewer found it necessary to give an explanation, we try to comment on the spatial difference between the central and west CLP revealed by Nie et al. (2008). We generally agree with the explanation given by Nie et al. (2008) and Guo et al. (2022). But we believe that the 100,000-year cyclic variation observed in the MaDang profile for MS is not a result of post-depositional pedogenic processes smoothing down, leading to the loss of the 20,000-year cyclic variation. A paragraph has been added to the revised manuscript to provide an explanation about this. *“Could both MS and FeD/FeT serve as proxies for precipitation in Madang? Previous studies have suggested that MS increased in the underlying materials and attenuated the precessional signal in CLP deposits (Nie et al., 2008). The MS and FeD/FeT records in the CLP show similar cycles, suggesting that smoothing has a similar influence on both proxies. In contrast, the MS and FeD/FeT records in Madang show different cycles, which would require precisely controlled differential smoothing to preserve the 20-kyr cycle of FeD/FeT but erase the 20- and 40-kyr cycles of MS. However, there is no evidence for this. It is likely that the MS and FeD/FeT of the Madang profile are related to different climate signals.”*

Commented [A24]: Formation mechanisms of hematite are currently in debate. It may not be a

good idea to put too much emphasis on hematite content variations.

Thanks for the suggestion. After weighing the options, we prefer to present previous studies objectively.

Commented [A25]: I would suggest comparing FeD/FeT with the magnetic monsoon proxy record from Tengger desert which also shows dominant precessional cycles. In doing so, the author team can examine monsoon variations from a more broad area and if they show consistent pattern, it would suggest the original interpretation for the MS record on the central CLP may have issues, whether it is postdepositional issues or temperature issue. This would be a more persuasive way. This is a constructive suggestion. The results in the monsoon marginal zone (Liu et al., 2021; Guo et al., 2022) were cited in the revised Discussion. Our result was compared with that from Tengger desert and the west CLP. A diagram was given to illustrate the well correlation between them (Fig. S13). The comparison shows that the higher insolation generally corresponds to the higher MAP or stronger EASM.

Fig. S13 Comparison of FeD/FeT record of Madang with a wet–dry record of Cahhanchi Lake in Tengger Desert (Liu et al., 2021), a spliced EASM record of Xijin drill cores on the western CLP (Guo et al., 2022), and the insolation difference between N30 and S30 in July (Laskar et al., 2004). The vertical bars indicate the intervals of low solar insolation, which generally correspond to low precipitation in both monsoon core zone (Madang) and monsoon marginal zone (Chahanchi Lake and Xijin)

Commented [A26]: I like this part. I think this is convincing no matter what MS variations indicate.
Thanks for recognition.

Commented [A27]: This is good and intriguing!
Thanks for recognition.

Commented [A28]: This does not belong to Results and also please consider my previous interpretations using postdepositional processes including bioturbation and leaching of iron across different layers to interpret lack of precessional cycles in Fe-related monsoon proxies. After all, not all sites on the loess plateau show dominant 100-kyr cycles and some do show apparent precessional cycles.

This sentence was deleted in revised manuscript, and the effects of postdepositional processes were presented as the possible cause of lack of precessional cycles in Fe-related monsoon proxies.

Commented [A29]: It would be useful to compare DARs between this site and the CLP (at least the Luochuan site) as leaching has larger impact for low DAR site. Western CLP sites have high DARs and are less susceptible to leaching than other sites. This had great impact on MS variations. Check the MS data comparison between the central and western CLP sites for the past three glacial-interglacial cycles (Zhang Jun et al, QSR): during the interglacials, the western sites have clear precessional cycles that central sites lack.

The mean DAR at Madang (5.25 cm/ka) is generally higher than that at Luochuan (4.40 cm/ka) during the past 346 ka. We also checked the DAR down the profile, and found that there is no abnormally low DAR near the four periods (~90 ka, ~160 ka, ~170ka, and ~248 ka, indicated by the orange dots in the following figure) where FeD/FeT is inconsistent with the \$\delta^{18}\text{O}\$ record. It seems that the potential leaching was not due to the low DAR.

Commented [A30]: I am convinced about this statement based on the evidence presented by the authors and their interpretation. This is an important findings from the EASM core region and this finding itself, lends me support publication of this manuscript, if the authors can make proper revisions to address the comments made, which are relatively stragiforward to take.

We thank the reviewer for his recognition. We acknowledge the necessity of revising

this manuscript to incorporate more recent study findings, including the records with a dominant ~20-kyr cycle in both lake sediments in Tengger Desert (Liu et al., 2021) and loess deposits in the western CLP (Guo et al., 2022). This will contribute to re-evaluating EASM cyclicities and dynamics across a broader region. We have thoroughly addressed all concerns raised by the reviewer and implemented necessary revisions. We trust that our modifications meet the reviewer's requirements.

Commented [A31]: There may have some unclear expression here: did you filter the FeD/FeT record and make the R calculation? Otherwise, these two curves can not have such high correlation coefficient because one is obviously have higher frequency variations that the other curve lacks. It is true that both series have undergone a filtering process prior to calculating the cross-correlation. In the Bincor package, each series is divided into multiple parts when two series have different timescale. The mean time and corresponding values of each part are then calculated for both series in order to standardize the timescale (Polanco-Martinez, et al., 2019). This filtering process has been applied to all cross-correlation calculations in this paper.

Polanco-Martinez, J. M. et al. BINCOR: An R package for estimating the correlation between two unevenly spaced time series. *The R Journal* 11, 170–184 (2019).

Commented [A32]: I agree that this work have merits because of the superior loess site location than other neighbouring sites. Otherwise, following the traditional way of research, one will not be able to generate this precious record from the EASM core region. Thanks for recognition.

Commented [A33]: Essentially because of the high elevation, right? We suspect that this is due not only to the absolute altitude, but also to the difference in altitude, as shown in the following sketch map. Here, we have added a paragraph to explain this issue. “The Madang profile is located in the piedmont zone along the Yangtze River, with a ground elevation exceeding 30 m. Simultaneously, the Yangtze River valley in Pengze County, where Madang is situated, runs in a northeast-southwest direction, aligning with the prevailing local wind direction (Fig. S1). Relatively high wind speeds, coupled with the profile's elevated position, contribute to the excellent ventilation and drainage conditions of the Madang profile.”

Fig. S1 Terrain in the lower reaches of the Yangtze River (a) and the adjacent Madang profile (b). The arrow indicates the prevailing wind direction. The purple rectangle in (a) indicates the area of (b).

Commented [A34]: See my comments before. Dr. Chen Fahu’s group generated a MS record from the western CLP which shows dominant precessional cycles. Based on this record, Guo et al. 2022 did the splicing. So the statement here is not precise and it neglects the basis for the splicing. Judging from Dr. Chen Fahu group’s MS data, I don’t think that you can argue that MS on the CLP reflects temperature control. So maybe you would need to use the suggestion that I made regarding how postdepositional processes changed the precessional cycles recorded by ultrafine ferromagnetic grains in soils to interpret why the central CLP MS does not have precessional cycles. The following sentences were deleted, “This approach may artificially suppress the generation of 100 kyr and 41 kyr cycles. It is evident that the Madang record constructed using a single indicator can more accurately reflect the true variations in EASM precipitation.” And the following paragraph was added. “Two records from the monsoon marginal zone show a cyclic variation similar to the East Asian speleothem record over the past ~300 kyr (Liu et al., 2021; Guo et al., 2022). A record of EASM precipitation from the western CLP shows a dominant cycle of 23 kyr over the past 260 kyr (Guo et al., 2022), while another record of wet–dry variations from the Tengger Desert has a clear 20-kyr cycle over the past 400 kyr (Liu et al., 2021). These are similar to the dominant 18-kyr cycle identified in this study, suggesting that precipitation (wetness) has varied synchronously in the monsoon zone, with high precipitation (wetness) generally correlated with high solar insolation (Fig. S13).”

Commented [A35]: How did you recognize these four patterns?

Four climatic modes can be seen in Fig. 4, for example, the warm-dry (~115 ka), warm-wet (~125 ka), cold-dry (~136 ka), and cold-wet (~147 ka).

Commented [A36]: What insights would we have considering this new record?

A paragraph was added to illustrate the insights. “Our results suggest that changes in

the East Asian summer monsoon, as indicated by precipitation in this region, are mainly forced by precession-dominated insolation changes. Additionally, the precipitation and temperature over the past ~350 ka in the low latitudes of East Asia may have varied with different cycles driven by different forcing factors. This scenario might also have occurred in a broader region, such as the CLP. However, more evidence is needed to verify this speculation. Our findings also indicate an urgent need to reassess the interpretation of proxies that are widely used in the CLP. Clearly, it is necessary to properly separate precipitation from temperature in geological records when trying to understand the nature of the EASM.”

Reviewer #3 (Remarks to the Author):

I think this manuscript has been improved greatly and I thank the author team for making revisions considering the comments made in last round of reviews. Particularly, they have adjusted title and statements in Abstract to de-emphasize link between magnetic susceptibility and temperatures. This paper could be accepted after minor revisions.

I don't have major issues with this version except one point. In this version, they attributed enhancement of magnetic susceptibility to oxidation of magnetite (line 196) which they think are temperature-controlled. However, I don't think the available references regarding magnetic susceptibility enhancement support this argument because oxidation of magnetite to maghemite would produce negligible changes in magnetic susceptibility and oxidation of magnetite to hematite would cause magnetic susceptibility decrease.

I still think that leaching-related signal smoothing hypothesis, as I proposed in my last round of reviews, can explain the dominant 100-kyr cycles in the magnetic susceptibility record in their study site. But they rejected this hypothesis in lines 190-193 because they think this factor would have affected magnetic susceptibility to the same degree, as is the case for the Chinese Loess Plateau, which was not observed in their site. Here I try to interpret my idea because I may have not made myself clear in my last round of reviews. Leaching-related smoothing can decrease power of precessional cycles, similar to the case in the Chinese Loess Plateau. However, as they demonstrated, climate in their site is superior than the Loess Plateau in producing free (CBD-dissolvable) iron. Therefore, free iron content is (likely much) larger than content of ultrafine ferrimagnetic grains (determining degree of magnetic enhancement) in their site, resulting in free iron/total iron proxy less affected by leaching-related signal smoothing. This is different from the Loess Plateau case where CBD-dissolvable iron oxides may be approximately equal to ultrafine ferrimagnetic minerals.

Therefore, my suggestion is to further de-emphasize the link between magnetic susceptibility and temperature (but it is still a good idea to mention this possibility) in the rest part (particularly lines 186-221 and this part belongs to Discussion in my opinion; lines 319-326) of the manuscript, and discuss the alternative interpretation (leaching-caused signal smoothing leaves different imprints on magnetic susceptibility and free iron/total iron). Revision of this nature would make the rest part of the manuscript match with the title and the abstract better in terms of how to explain the cyclicities in their magnetic susceptibility record, and it would allow them to introduce balanced hypotheses regarding how to explain different cyclicities in the magnetic susceptibility and the free iron/total iron records in their site.

I feel that the suggested revisions are quite straightforward to make and I emphasize that these additional revisions would not jeopardize the novelty and robustness of this research, which allows a glimpse of orbital-timescale monsoon variations from the monsoon key area. Also, these revisions would put their conclusions on a safer ground no matter whether future studies support the potential link between magnetic susceptibility and temperature in southern China soils or not.

RESPONSE TO REVIEWERS (highlighted in blue)

REVIEWERS' COMMENTS

Reviewer #3 (Remarks to the Author):

I think this manuscript has been improved greatly and I thank the author team for making revisions considering the comments made in last round of reviews. Particularly, they have adjusted title and statements in Abstract to de-emphasize link between magnetic susceptibility and temperatures. This paper could be accepted after minor revisions.

I don't have major issues with this version except one point. In this version, they attributed enhancement of magnetic susceptibility to oxidation of magnetite (line 196) which they think are temperature-controlled. However, I don't think the available references regarding magnetic susceptibility enhancement support this argument because oxidation of magnetite to maghemite would produce negligible changes in magnetic susceptibility and oxidation of magnetite to hematite would cause magnetic susceptibility decrease.

I still think that leaching-related signal smoothing hypothesis, as I proposed in my last round of reviews, can explain the dominant 100-kyr cycles in the magnetic susceptibility record in their study site. But they rejected this hypothesis in lines 190-193 because they think this factor would have affected magnetic susceptibility to the same degree, as is the case for the Chinese Loess Plateau, which was not observed in their site. Here I try to interpret my idea because I may have not made myself clear in my last round of reviews. Leaching-related smoothing can decrease power of precessional cycles, similar to the case in the Chinese Loess Plateau. However, as they demonstrated, climate in their site is superior than the Loess Plateau in producing free (CBD-dissolvable) iron. Therefore, free iron content is (likely much) larger than content of ultrafine ferrimagnetic grains (determining degree of magnetic enhancement) in their site, resulting in free iron/total iron proxy less affected by leaching-related signal smoothing. This is different from the Loess Plateau case where CBD-dissolvable iron oxides may be approximately equal to ultrafine ferrimagnetic minerals.

Therefore, my suggestion is to further de-emphasize the link between magnetic susceptibility and temperature (but it is still a good idea to mention this possibility) in the rest part (particularly lines 186-221 and this part belongs to Discussion in my opinion; lines 319-326) of the manuscript, and discuss the alternative interpretation (leaching-caused signal smoothing leaves different imprints on magnetic susceptibility and free iron/total iron). Revision of this

nature would make the rest part of the manuscript match with the title and the abstract better in terms of how to explain the cyclicities in their magnetic susceptibility record, and it would allow them to introduce balanced hypotheses regarding how to explain different cyclicities in the magnetic susceptibility and the free iron/total iron records in their site.

I feel that the suggested revisions are quite straightforward to make and I emphasize that these additional revisions would not jeopardize the novelty and robustness of this research, which allows a glimpse of orbital-timescale monsoon variations from the monsoon key area. Also, these revisions would put their conclusions on a safer ground no matter whether future studies support the potential link between magnetic susceptibility and temperature in southern China soils or not.

We appreciate the reviewer's acknowledgement and encouragement of our last round of revisions. Following the reviewer's suggestions, we have further refined the manuscript. The main changes mainly involve a further reduction in the discussion of the relationship between magnetic susceptibility and temperature. This adjustment aims to achieve a balanced structure of the paper, while emphasizing the findings and the significance of the 20-kyr monsoon precipitation cycle forced by precession variation.

We appreciate the reviewer's recognition and encouragement for our last round of revisions. Following the reviewer's suggestions, we have further revised the manuscript. The new modifications mainly involve a further attenuation the link between magnetic susceptibility and temperature, aiming to balance the structure of the paper and emphasize the finding and significance of the 20-kyr monsoon precipitation cycle.

Major revisions include:

(1) We introduce an alternative hypothesis for the MS enhancement, namely precipitation-induced signal smoothing. “Could both MS and FeD/FeT serve as proxies for precipitation in Madang? The existing evidence is insufficient to give a definite answer, but two hypotheses can be proposed: (1) Both MS and FeD/FeT serve as proxies for precipitation; (2) MS and FeD/FeT have different climatic indications.

The first hypothesis involves post-depositional processes. Previous studies have suggested that leaching-related smoothing can diminish the strength of precessional cycles. Pedogenesis during interglacial periods can penetrate to greater depths, increasing the MS in underlying materials and attenuating the precessional signal in CLP loess deposits (Nie et al., 2008). The MS and FeD/FeT records in the CLP show similar cycles, suggesting that smoothing has a similar influence on both proxies. In contrast, the MS and FeD/FeT records in Madang show different cycles, which would

require precisely controlled differential smoothing to preserve the 20-kyr cycle of FeD/FeT but erase the 20-kyr cycle of MS. One possible interpretation is that the climatic conditions at the Madang site favour the production of FeD compared to the CLP (Yang et al., 2022). Consequently, the FeD content at the Madang site significantly exceeds the concentration of ultrafine ferrimagnetic grains, which determines the degree of magnetic enhancement. As a result, the FeD/FeT proxy at the Madang site may be less influenced by leaching-induced signal smoothing, unlike the CLP site where DCB-soluble iron oxides may be roughly equivalent to ultrafine ferrimagnetic minerals.”

(2) We replace “oxidation of ultrafine magnetite grains” with “soil moisture-induced redox status” for the enhancement mechanism of MS. “The second hypothesis involves the influence of temperature. Precipitation may not be a controlling factor for MS enhancement in wetter low latitudes. For example, the MS in Madang is generally lower than that in Luochuan. Excessive precipitation may suppress the enhancement of MS, as the MS of modern soils in southern China decreases with increasing precipitation (Han et al., 1996). A study has revealed an anomalous decrease in MS in the Xiashu loess, attributed to a strong reducing action caused by excessive soil moisture (Han et al., 2008). The soil moisture will decrease with enhanced evapotranspiration, which mainly increases with temperature. In this sense, MS enhancement at the Madang site may be more sensitive to temperature, resulting in variations in redox status”.

(3) We have removed the main content of lines 186-221 of the previous draft as mentioned by the reviewer, including the comparison of magnetic susceptibility with haematite, redness, and the reconstructed paleotemperature results based on GDGTs.

(4) To emphasise the finding and significance of the 20-kyr monsoon precipitation cycle, we have moved Figure 5 from the Supplementary Information (SI) to the main body of the paper. In addition, we have deleted the contents of lines 319-326 in the previous draft, as mentioned by the reviewer. We have also added a summary paragraph at the end to highlight the importance of this study. “In summary, the Madang profile in this study is exceptionally rare and valuable, constituting a high-resolution loess record from the core zone of the EASM. It affords a unique opportunity to reassess the cyclicity and dynamics of the EASM across a broad region based on aeolian dust deposits. Our results suggest that changes in the EASM, as indicated by precipitation in this region, are mainly forced by precession-dominated insolation changes, which addresses the key aspect of the “Chinese 100 kyr problem”.